# Tumor-specific lncRNA *IGF1R-AS1* trans-regulates chromatin interactions associated with oncogenic MYC signaling

Yongyong Yang[1,13], Ting-You Wang [1,13], Joshua Fry [1,2,13], Yingming Li[3,13], Qingshu Meng [1,13], Qingxiang Guo [1], Nathan E. Patchen[4], Kyle H. White [5,12], Abhirami Ramakrishnan [1], Yanan Ren[1], Qianru Li[6], Xingxing Zhang[1], Taufeeque Ali[1], Courtney Dawes [4], Stamatina Fragkogianni[7], Parker Irvin[5], Sk Kayum Alam [8], Luke H. Hoeppner [3,8], Xihong Zhang[3], Douglas Yee [3], Adam B. Weiner[9], Edward M. Schaeffer [1], Yang Liu [4], Xiaoyang Zhang[5], Scott M. Dehm [3,10] ✉, Qi Cao [1,11] ✉ & Rendong Yang [1,11] ✉

LncRNAs have emerged as pivotal regulators in the development and progression of various human cancers. However, understanding the precise mechanisms by which lncRNAs influence cancer progression remains a substantial challenge, largely due to their cell type- and tissue-specific expression patterns and the lack of well-defined functional domains or motifs. In this study, we investigate the complex interplay between super-enhancers and lncRNAs through a comprehensive analysis of lncRNA expression in a cohort of metastatic castration-resistant prostate cancer patients. Our analysis identifies 1344 lncRNAs, among which an antisense lncRNA in the *IGF1R* locus named *IGF1R-AS1* displayed the strongest super-enhancer association. Through pan-cancer transcriptome analysis, we find that *IGF1R-AS1* is specifically transcribed in tumor specimens and is overexpressed in prostate and lung cancers. Notably, we reveal a non-canonical *trans*-acting role for *IGF1R-AS1* whereby it interacts with chromatin remodeling complexes and architectural proteins to facilitate long-range chromatin looping between distal *MYC* enhancers and its promoter, leading to *MYC* overexpression and enhanced tumorigenicity. Collectively, our findings elucidate a mechanism by which a tumor-specific *trans*-acting lncRNA modulates oncogenic *MYC* expression through long-range chromatin interactions, suggesting *IGF1R-AS1* may play an important role in the pathogenesis of MYC-driven malignancies.

Long non-coding RNAs (lncRNAs) play critical roles in the regulation of gene expression, particularly in cancer progression and lineage-specific oncogenic programs[1]. While high-throughput transcriptomic studies have uncovered thousands of lncRNAs, their context-specific functions and regulatory mechanisms remain incompletely understood. Many well-characterized cancer-associated lncRNAs, such as

*MALAT1*, *HOTAIR*, and *NEAT1*, are broadly expressed across both normal and tumor tissues[2]. In contrast, some lncRNAs exhibit highly tissue- or tumor-specific expression patterns and tend to act locally (in *cis*), especially at super-enhancer regions where they can promote chromatin looping and transcriptional activation of nearby oncogenes[3]. For example, the colorectal cancer-associated lncRNA

*CCAT1*, transcribed from a super-enhancer upstream of *MYC*, facilitates enhancer-promoter interactions that drive *MYC* overexpression[4].

Despite such advances, the landscape of lncRNAs with tumor-specific expression, those absent in normal tissues but selectively activated in cancer, remains poorly characterized. This class of lncRNAs holds particular promise for cancer-specific targeting but is often missed in annotation databases due to their selective expression in tumor contexts[5]. A deeper understanding of tumor-specific lncRNAs and their mechanisms of action may offer unique opportunities for therapeutic intervention and disease monitoring.

While certain tumor-specific lncRNAs with *cis*-regulatory functions have been well characterized, the potential for tumor-specific lncRNAs to act in *trans*, regulating gene expression at distant genomic loci, remains less well understood. A particularly open question is whether *trans*-acting lncRNAs can modulate 3D chromatin architecture in a manner similar to their *cis*-acting counterparts. Although some lncRNAs have been shown to scaffold transcriptional regulators or influence RNA stability[6], direct evidence of their role in mediating long-range chromatin interactions across genomic loci, especially in *trans*, is still limited.

In this study, we identify and characterize *IGF1R-AS1*, an antisense lncRNA with remarkably specific expression in prostate cancer and a subset of lung tumors. Unlike most lncRNAs that show expression across both normal and tumor tissues, *IGF1R-AS1* is virtually absent from normal samples and significantly enriched in advanced disease, suggesting a tumor-specific and potentially oncogenic role. Functional analyses demonstrate that *IGF1R-AS1* promotes cancer cell proliferation, migration, and survival, both in vitro and in xenograft models. Notably, *IGF1R-AS1* depletion leads to downregulation of *MYC* and disruption of enhancer-promoter interactions at the *MYC* locus,

without affecting *IGF1R* expression in *cis*. While the complete mechanism requires further investigation, our data supports a model where *IGF1R-AS1* mediates long-range chromatin interactions between multiple *cis*-regulatory enhancer elements and the *MYC* promoter. These findings not only expand the catalog of tumor-specific lncRNAs but also reveal a mechanism through which non-coding RNAs can influence distal gene regulation and 3D genome architecture in cancer.

## Results

### Discovery of lncRNAs from transcriptome analysis of mCRPC

Understanding lncRNA-mediated transcriptional regulation in human cancers, particularly those lacking effective treatments, remains a critical research priority. To gain deeper insight into these regulatory mechanisms, we focused on metastatic castration-resistant prostate cancer (mCRPC), a lethal form of cancer, as our model system. We performed a meta-assembly of mCRPC transcriptomes by analyzing 101 poly-A purified RNA-seq samples obtained from the AACR-PCF Stand-Up-To-Cancer (SU2C) study[7] (See "Methods" for details). This comprehensive analysis enabled us to detect recurrently expressed lncRNA transcripts in these mCRPC samples. We identified a total of 1344 unannotated lncRNAs against the GENCODE gene annotation[8] that were expressed in at least 5 of 101 (5%) mCRPC samples (Fig. 1a and Supplementary Data 1). These unannotated lncRNAs were evolutionarily less conserved than protein-coding genes but were comparable to annotated lncRNAs (Fig. 1b). To ensure that these unannotated lncRNA candidates did not encode proteins, we considered assembled transcripts as lncRNAs when they were predicted to have no coding potential by five widely used tools (See "Methods" for details). We observed the unannotated transcripts have a lower coding potential relative to protein-coding genes but similar coding potential to

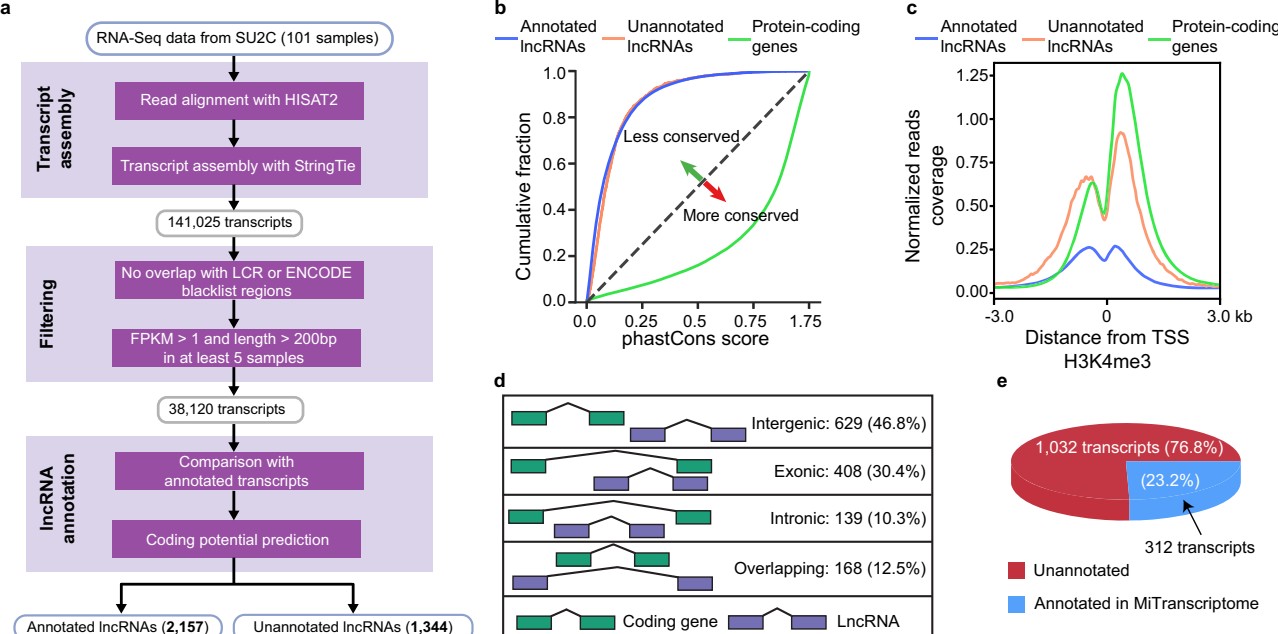

**Fig. 1 | LncRNA landscape in mCRPC. a** Computational workflow of annotated and unannotated lncRNA discovery. SU2C Stand Up to Cancer's cohort, LCR Low Complexity Region. **b** Cumulative distribution of phastCons scores. The plot compares conservation levels of annotated lncRNAs (blue), unannotated lncRNAs (orange), and protein-coding genes (green). Both annotated and unannotated lncRNAs show low conservation, while protein-coding genes are highly conserved. The dashed diagonal line indicates random conservation, with arrows marking less conserved (green) and more conserved (red) regions. **c** H3K4me3 histone modifications at transcription start sites (TSS) in LNCaP cells. Normalized read coverage of H3K4me3 marks is shown for annotated lncRNAs (blue), unannotated lncRNAs

(orange), and protein-coding genes (green). H3K4me3 enrichment is strongest at the TSS of protein-coding genes, followed by unannotated and annotated lncRNAs, indicating differences in promoter activity. **d** Positional classification of unannotated lncRNAs relative to nearby protein-coding genes. Unannotated lncRNAs are classified based on their genomic location relative to protein-coding genes: intergenic (46.8%), exonic (30.4%), intronic (10.3%), and overlapping (12.5%). Green boxes represent coding genes, and purple boxes represent lncRNAs. The schematic illustrates the positional relationships, highlighting the diverse genomic contexts of these lncRNAs. **e** Annotations of unannotated lncRNAs by MiTranscriptome database.

annotated lncRNAs (Supplementary Fig. 1a). In addition, these unannotated lncRNAs tended to have fewer exons and shorter transcript lengths than protein-coding genes (Supplementary Fig. 1b, c).

To further validate the expression of these unannotated lncRNAs, we leveraged the publicly available ENCODE ChIP-seq data of epigenetic signatures (H3K4me3, H3K27ac, Pol II binding) that are associated with active transcription start sites. We observed the H3K4me3 histone modification, an active promoter marker, was highly enriched at the predicted start sites of these lncRNAs (Fig. 1c). Moreover, these lncRNAs showed greater H3K4me3 enrichment in prostate cancer cells compared to other cancer types (Supplementary Fig. 1d), suggesting their potential association with prostate cancer. We further validated the transcription of these unannotated lncRNAs by analyzing RNA Pol II and H3K27ac ChIP-seq signals (Supplementary Fig. 1e, f) and by examining aggregate RNA-seq read density along the full-length of all these lncRNAs (Supplementary Fig. 1g) in VCaP prostate cancer cells.

We adapted criteria from a previous study[9] to classify these unannotated lncRNAs into four categories, which revealed that most of them are intergenic relative to protein-coding genes (Fig. 1d). In addition, we compared these unannotated lncRNAs with those reported by the MiTranscriptome study[10], which is a catalog of human lncRNAs derived from computational analysis of RNA-seq data from over 6500 samples spanning diverse cancer and tissue types. This comparison revealed that fewer than a quarter of these unannotated lncRNAs were captured by MiTranscriptome (Fig. 1e).

## Identification of mCRPC-associated lncRNA *IGF1R-AS1* through super-enhancer analysis

To investigate the functions of these unannotated lncRNAs in mCRPC, we compared RNA-seq data from our mCRPC cohort with RNA-seq data that were generated from normal prostate ($n = 52$) and localized prostate cancer ($n = 500$) by The Cancer Genome Atlas (TCGA) study[11]. We focused on the lncRNAs differentially expressed in mCRPC compared to normal prostate tissues, which revealed annotated mCRPC-associated lncRNAs as well as unannotated lncRNA candidates that were primarily highly expressed in mCRPC (Fig. 2a). Next, we sought to understand the potential functions of these unannotated lncRNAs by linking their expression to super-enhancers (SEs) in VCaP cells, which were identified by assessing the H3K27ac ChIP-seq density with the ROSE algorithm[12]. This analysis revealed SEs overlapping known prostate cancer protein-coding genes and lncRNAs (e.g., *AR*, *TMPRSS2*, *FOXA1*, *ERG*, *PCAT2*, *NEAT1*, *PCAT1*), as well as a subset of unannotated lncRNAs we had identified ($n = 111$) (Fig. 2b). Among them, we focused on the highest-ranked SE-associated unannotated lncRNA, an anti-sense transcript located within the intronic region of *IGF1R* gene, hereafter named *IGF1R-AS1* (Fig. 2b). *IGF1R-AS1* was one of the most highly expressed SE-overlapping genes in comparisons of localized prostate cancer vs. benign tissue and mCRPC vs. localized prostate cancer (Fig. 2c), nominating it as an unannotated lncRNA for further investigation.

We noted that GENCODE annotated two antisense transcripts (*ENSG00000287191*) around the region where *IGF1R-AS1* is located. However, these annotated transcripts displayed isoform structures distinct from the *IGF1R-AS1* isoform predicted by StringTie (Fig. 2d). To address this discrepancy, we performed 3′ rapid amplification of cDNA ends (3′RACE) using a forward primer targeting exon 1 of *IGF1R-AS1*. We first conducted 3′RACE reactions with RNA from VCaP cells, which express high levels of *IGF1R-AS1* (Supplementary Fig. 2a), and subjected the resulting 3′RACE products to PacBio Iso-seq long-read sequencing. This approach revealed several full-length *IGF1R-AS1* isoforms (Supplementary Table 1), most of which shared a splicing junction between exon 1 and exon 2, with splice donor and acceptor sites consistent with those predicted by StringTie (Fig. 2d). We also analyzed the Nanopore direct RNA sequencing data in VCaP cells[13] and

confirmed the consistency of *IGF1R-AS1* integrity and size (Supplementary Fig. 2b). Furthermore, we observed that the major *IGF1R-AS1* isoforms identified in VCaP cells were also expressed in the lung cancer cell line H727, another cancer cell line with high endogenous expression of *IGF1R-AS1* (Supplementary Fig. 2c and Supplementary Table 1). Collectively, these findings confirm that *IGF1R-AS1* transcripts are expressed in multiple cancer cell lines.

To further explore the association of *IGF1R-AS1* with SEs, we examined the H3K27ac ChIP-seq data of VCaP cells and identified a 20 kb SE downstream of *IGF1R-AS1*, co-occupied by RNA Pol II and the transcriptional co-activator BRD4 (Fig. 2e). Furthermore, RNA Pol II ChIA-PET data from VCaP cells revealed direct interactions between this SE and the enhancer upstream of *IGF1R-AS1*, marked by H3K4me3 and H3K27ac modifications (Fig. 2e). However, this SE-enhancer interaction was not detected in LNCaP and DU145 cells based on RNA Pol II ChIA-PET data (Supplementary Fig. 2d), which may explain the absence of *IGF1R-AS1* expression in these cells, as indicated by RNA-seq measurements (Supplementary Fig. 2a). To further confirm whether this SE induces *IGF1R-AS1* expression, we treated VCaP cells with bromodomain inhibitor JQ1 to reduce the BRD4 occupancy in this SE region (Fig. 2f), thereby selectively disrupting the expression of genes associated with active SEs. Notably, JQ1 treatment led to a significant reduction in the expression levels of both *IGF1R-AS1* and *MYC*, a well-characterized SE-associated gene, while *IGF1R* expression remained unaffected (Fig. 2g). Additionally, we analyzed existing RNA-seq data from JQ1-treated and untreated H727 lung cancer cells[14]. Consistent with our findings in VCaP cells, *IGF1R-AS1* expression decreased in JQ1-treated H727 cells, whereas *IGF1R* expression remained stable (Supplementary Fig. 2e). This result was further validated by qPCR analysis (Supplementary Fig. 2f).

Next, we sought to identify transcription factors (TFs) targeting the *IGF1R-AS1*-associated SE to understand the regulatory mechanisms underlying its expression. Analysis of publicly available ChIP-seq binding sites revealed enrichment of androgen receptor (AR) and its co-factors (FOXA1 and HOXB13) in this SE region (Supplementary Fig. 2g). We confirmed the involvement of AR as a positive regulator of *IGF1R-AS1* expression by suppressing AR with the antagonist enzalutamide and stimulating AR with the agonist dihydrotestosterone (DHT). However, we observed similar responses of *IGF1R* to these manipulations as *IGF1R-AS1* (Supplementary Fig. 2h, i), suggesting that regulation of *IGF1R-AS1* by AR is not exclusively mediated through this specific *IGF1R-AS1*-associated SE. Therefore, we investigated the next top-ranked TF, SMARCA4 (Supplementary Fig. 2g), a chromatin remodeler known to interact with AR[15]. We confirmed SMARCA4 binding to the *IGF1R-AS1*-associated SE in VCaP cells, while no binding was detected in the 22RV1 and LNCaP ChIP-seq data (Fig. 2h). This suggests that the hyper-expression of *IGF1R-AS1* in VCaP cells may indeed be attributed to SWI/SNF-mediated SE regulation. To further validate this hypothesis, we examined the chromatin accessibility at the *IGF1R-AS1*-associated SE and its distal enhancer using publicly available ATAC-seq data. Our results showed that SMARCA4 depletion disrupted chromatin accessibility specifically at the *IGF1R-AS1*-associated SE (Fig. 2h). Consequently, SMARCA4 inactivation markedly disrupted the three-dimensional interactions between the *IGF1R-AS1*-associated SE and its distal enhancer (Fig. 2i), as shown by the analysis of a published H3K27ac HiChIP dataset upon treatment with a PROTAC SWI/SNF degrader (AU-15330)[16]. Additionally, ATAC-seq analysis revealed diminished accessibility at the *IGF1R-AS1* promoter (Fig. 2i). This observation aligns with the reduction in *IGF1R-AS1* expression upon SMARCA4 degradation (Supplementary Fig. 2j) or depletion, whereas *IGF1R* RNA and protein levels remained relatively stable in both VCaP and H727 cells (Supplementary Fig. 2k–n). These data collectively support a model in which *IGF1R-AS1* expression is influenced by SWI/SNF complex-mediated SE regulation.

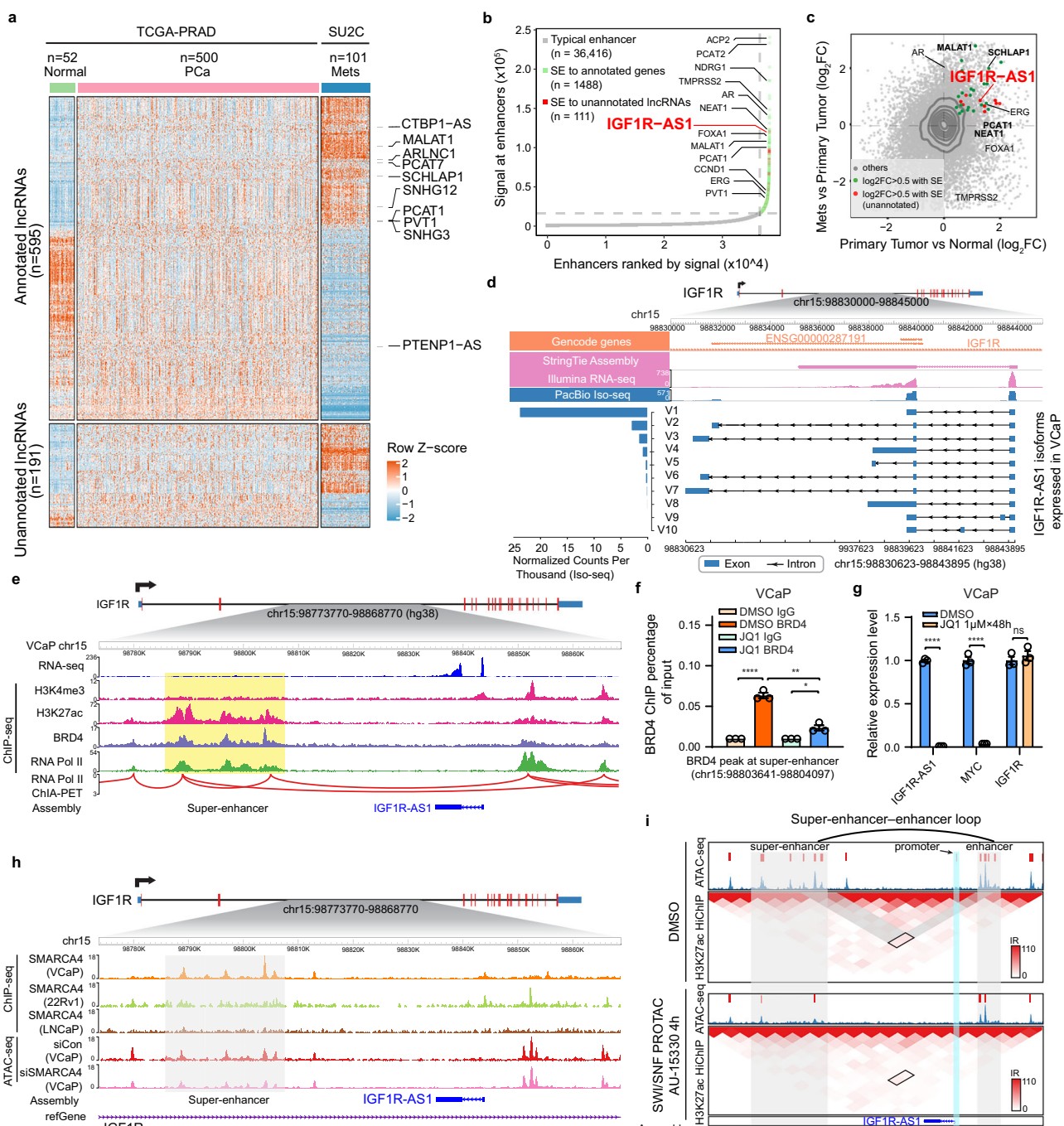

**Fig. 2 | Identification of lncRNA *IGF1R-AS1*. a** Heatmap of differentially expressed annotated (*n* = 595) and unannotated (*n* = 191) lncRNAs across benign prostate tissue (Normal, *n* = 52), localized prostate cancer (PCa, *n* = 500), and metastatic prostate cancer (Mets, *n* = 101) samples. RNA-seq data for Normal and PCa samples were obtained from the TCGA cohort, while Mets samples were sourced from the SU2C study. Rows represent lncRNAs, with annotated lncRNAs in the upper panel and unannotated lncRNAs in the lower panel. Columns represent individual samples grouped by disease state. Expression values are Z-score normalized across samples. Representative lncRNAs with established roles in prostate cancer progression (e.g., *CTBP1-AS*, *MALAT1*, and *SChLAP1*) are highlighted. **b** Super-enhancer-associated lncRNAs and protein-coding genes in VCaP cells based on ranked ordered H3K27ac ChIP-seq signal. **c** Identification of super-enhancer-associated coding genes and lncRNA transcripts elevated during prostate cancer progression. **d** Detected *IGF1R-AS1* isoforms by short- and long-read sequencing approaches. **e** Integrated genome view of *IGF1R-AS1* and its associated super-enhancer in VCaP cells. The data tracks represent RNA-Seq; ChIP-Seq for H3K4me3, H3K27ac, and RNA Pol II; ChIA-PET for RNA Pol II. **f** ChIP-qPCR

measurement of BRD4 binding at the *IGF1R-AS1*-associated super-enhancer region in JQ1-treated VCaP cells and DMSO control cells (*n* = 3 biological replicates). Unpaired Student's *t*-test, two-sided. *p*\*=0.0161, *p*\*\*=0.0011, *p*\*\*\*\*<0.0001. **g** RNA expression levels of *IGF1R-AS1*, *IGF1R*, and the known super-enhancer target gene *MYC* measured in VCaP cells treated with JQ1 and DMSO (*n* = 3 biological replicates). Unpaired Student's *t*-test, two-sided. *p*\*\*\*\*<0.0001, ns not significant. **h** SMARCA4 ChIP-seq binding in the *IGF1R-AS1*-associated super-enhancer across different prostate cancer cell lines and ATAC-seq signals under *SMARCA4* depletion and control VCaP cells at this super-enhancer region. **i** H3K27ac HiChIP heatmap within the *IGF1R-AS1* locus in VCaP cells treated with or without the SWI/SNF PROTAC degrader AU-15330 for 4 h (bin size = 5 kb). Overlaid on the heatmap are ATAC-seq read-density tracks from the same treatment conditions. Gray shading indicates typical- and super-enhancers, while blue shading marks the *IGF1R-AS1* promoter. The loop represents a read-supported *cis*-interaction within the locus, with interaction reads (IR) denoting the strength of this interaction. Data in (**f**, **g**) shown are mean ± SEM. Source data of this Figure are provided as Source data file.

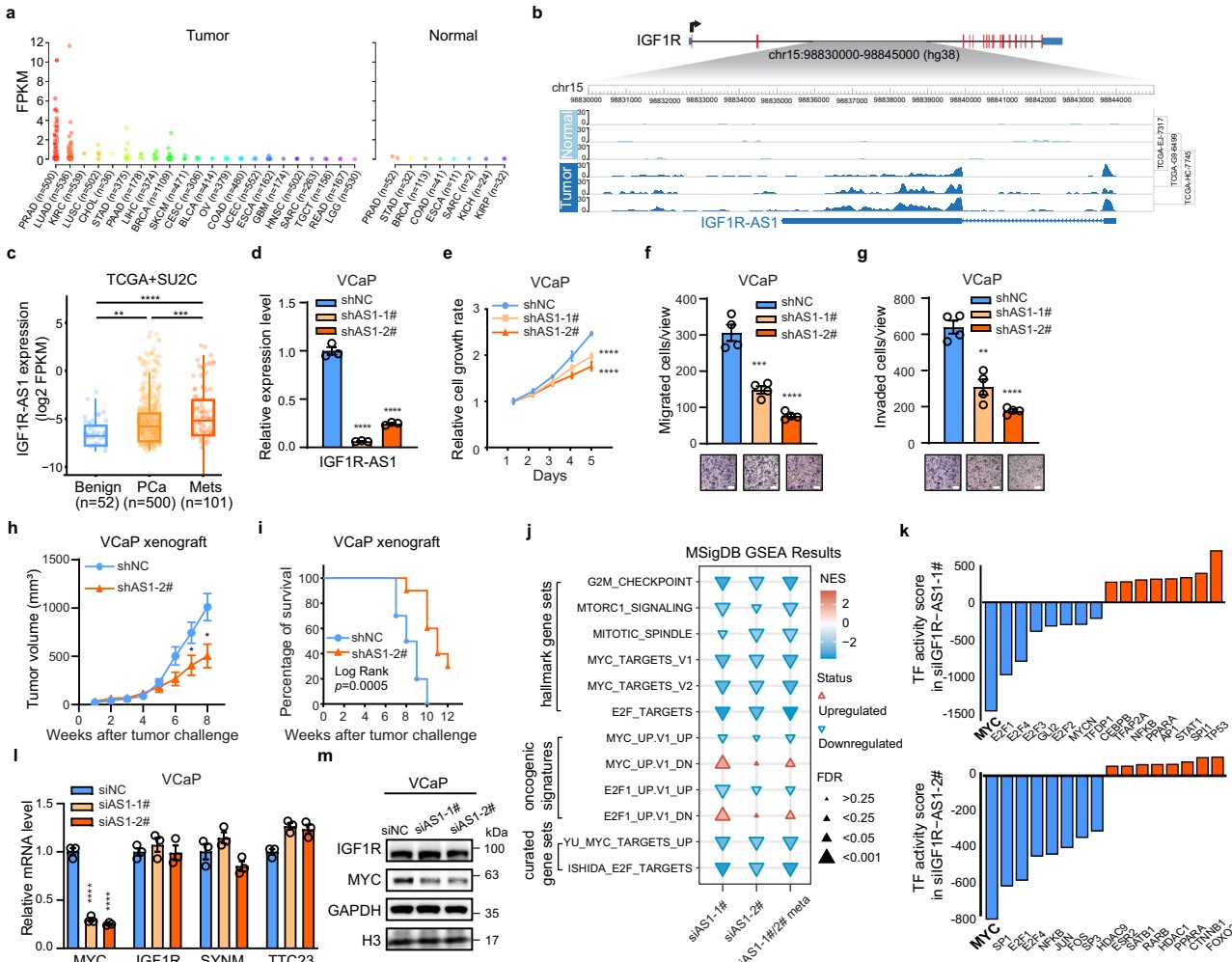

**Fig. 3 | *IGF1R-AS1* functions as an oncogenic lncRNA and activates MYC signaling. a** *IGF1R-AS1* expression across TCGA tumors and normal tissues. **b** RNA-seq coverage tracks from three TCGA patient samples showing *IGF1R-AS1* expression in prostate cancer (dark blue) and matched adjacent normal tissues (light blue). **c** *IGF1R-AS1* expression in benign, primary PCa, and metastatic (Mets) samples from TCGA-PRAD and SU2C cohorts. *n* is the patient sample number. Boxplots are shown as the median (center line), interquartile range (box bounds, 25th to 75th percentile), and whiskers extending to 1.5× the interquartile range, and points beyond the whiskers represent outliers. Unpaired Student's *t*-test, two-sided. *p***(Benign vs PCa) = 0.0022, *p*** (PCa vs Mets) = 0.0003, *p*****(Benign vs Mets) < 0.0001. **d** qPCR analysis of *IGF1R-AS1* knockdown in VCaP cells (*n* = 3 biological replicates). Unpaired Student's *t*-test, two-sided. *p*****<0.0001. **e–g** Depleting *IGF1R-AS1* inhibited cell growth (**e**, *n* = 6 biological replicates. *p*****(day 5) < 0.0001), migration (**f**, *n* = 4 biological replicates. p***=0.0007, *p*****<0.0001), and invasion (**g**, *n* = 4 biological replicates. p**=0.0010, *p*****<0.0001) of VCaP cells. Scale bar in (**f** and **g**), 200 μm. Unpaired Student's *t*-test, two-sided. **h** Tumor growth curve in xenograft mouse models of VCaP cells after *IGF1R-AS1* stable knockdown. *n* = 10 mice per group. Unpaired Student's *t*-test, two-sided. *p*\* (week 7) = 0.0374, *p*\* (week 8) = 0.0148. **i** Kaplan–Meier survival plot in xenograft mouse models of VCaP cells

after *IGF1R-AS1* stable knockdown, based on the weeks needed when tumor size reaches 1000 mm³. *n* = 10 mice per group. Log Rank test. **j** Gene set enrichment analysis (GSEA) of the *IGF1R-AS1*-related pathways in *IGF1R-AS1* knockdown VCaP cells. The normalized enrichment scores (NES) for each gene set are displayed across different experimental conditions, with blue indicating downregulation and red indicating upregulation. "siAS-1 #1/#2 meta" refers to a GSEA performed on the results of a meta-analysis of differentially expressed genes (DEGs) from independent siAS-1 #1 and siAS-1 #2 knockdown experiments (*n* = 3 biological replicates). **k** Transcription factor (TF) activity analysis under *IGF1R-AS1* knockdown conditions was performed using TF networks from the CollecTRI database. Bar plots display the top 8 upregulated TFs (orange) and bottom 8 downregulated TFs (blue), ranked by activity scores inferred from RNA-seq data following *IGF1R-AS1* knockdown with two independent siRNAs in VCaP cells. **l, m** qPCR analysis (**l**, *n* = 3 biological replicates) and western blotting analysis (**m**) of the expression of *MYC*, *IGF1R*, and *IGF1R* nearby genes, *SYNM* and *TTC23*, in VCaP cells after *IGF1R-AS1* knockdown. Unpaired Student's *t*-test in (**l**), two-sided. *p*****<0.0001. Data in (**d–h**, and **l**) are shown as mean ± SEM. Experiments in (**m**) were biologically repeated three times. Source data of this Figure are provided as Source data file.

## Cancer-specific *IGF1R-AS1* functions in *trans* to upregulate MYC signaling independently of IGF1R

We hypothesized that the tissue- and disease-specific expression of lncRNAs are the basis for the absence of certain lncRNAs, such as *IGF1R-AS1*, from lncRNA databases. To investigate this further, we conducted a comprehensive analysis of *IGF1R-AS1* expression across multiple tissues using RNA-seq data from the TCGA pan-cancer cohort. We compiled data from 9057 RNA-seq samples across 22 cancer types, comprising 8750 tumor samples and 307 normal samples. Our analysis

revealed that *IGF1R-AS1* exhibited pronounced specificity for prostate tissue and cancer, while also showing elevated expression in lung adenocarcinoma compared to other cancer types (Fig. 3a). Moreover, we examined the top three *IGF1R-AS1*-high TCGA prostate specimens for which tumor and matched adjacent normal tissue RNA-seq data were available. This tumor/normal matched comparison further confirmed the cancer-specific expression of *IGF1R-AS1* (Fig. 3b), indicating its potential oncogenic functions. Consistent with this observation, we found that *IGF1R-AS1* expression was significantly higher in localized

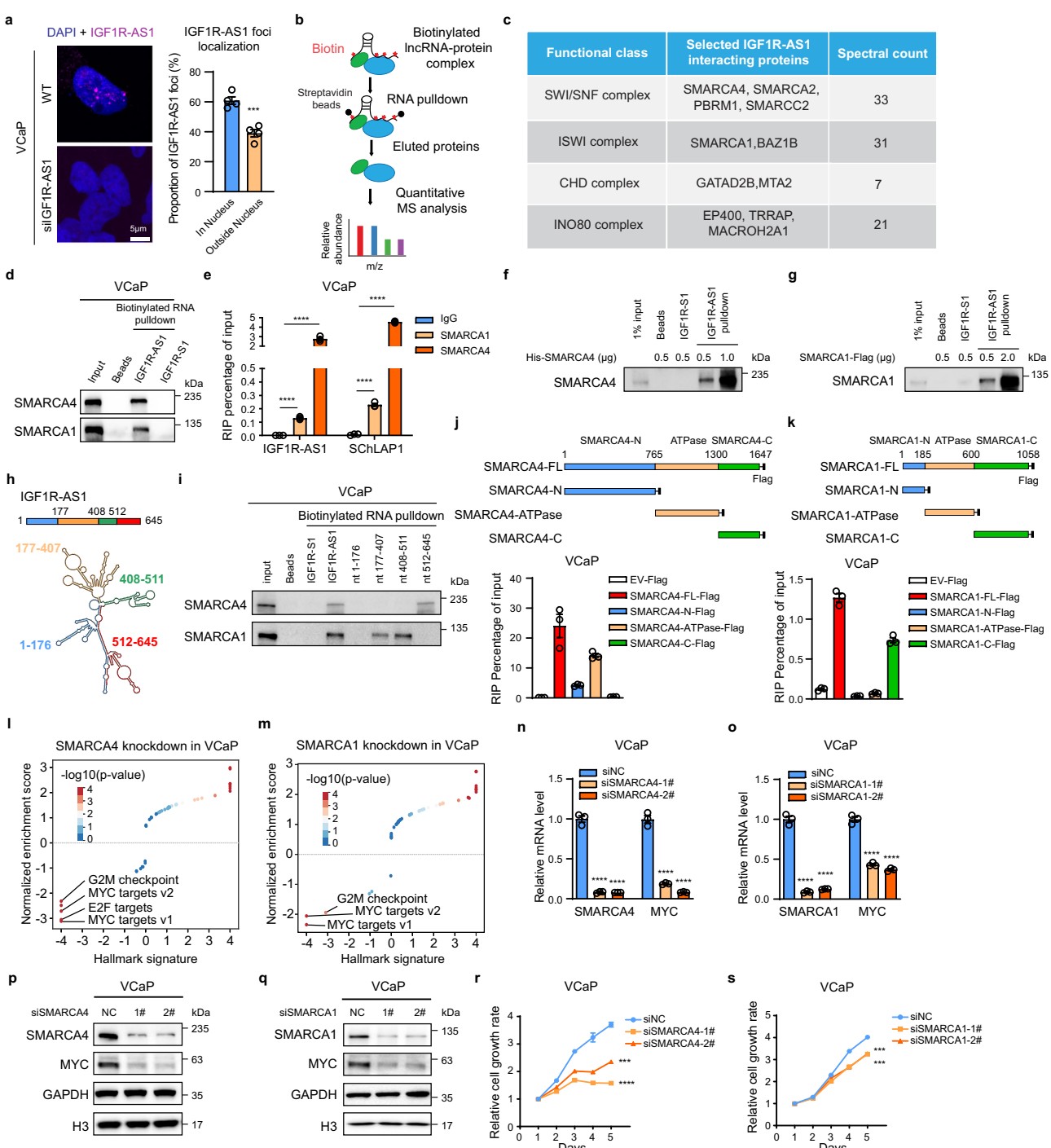

and metastatic tumors compared to benign tissues (Fig. 3c). Assessment of the *IGF1R-AS1* feature junction revealed increased prevalence in metastatic compared with localized tumors (Supplementary Fig. 3a).

To validate *IGF1R-AS1*'s oncogenic potential, we knocked down *IGF1R-AS1* expression through RNA interference (shRNAs or siRNAs) in VCaP, 22RV1, and H727 cells (Fig. 3d and Supplementary Fig. 3b). Cell growth rates of VCaP, 22RV1, and H727 were decreased in *IGF1R-AS1* knockdown cells compared to control cells (Fig. 3e and Supplementary Fig. 3c). Both cell migration and invasion were significantly reduced in *IGF1R-AS1* knockdown cells compared to those in control cells (Fig. 3f, g and Supplementary Fig. 3d). Restoring *IGF1R-AS1* expression in VCaP knockdown cells reversed the inhibited cell growth, migration, and invasion (Supplementary Fig. 3e–g). Overexpressing *IGF1R-AS1* in low-expressing LNCaP cells resulted in a moderate increase in cell growth,

migration, and invasion (Supplementary Fig. 3h–j). We then investigated whether *IGF1R-AS1* affects the growth of VCaP cells grown as xenografts in immunocompromised mice. Growth of the *IGF1R-AS1*-knockdown xenografts was significantly suppressed compared to the growth of control xenografts (Fig. 3h), resulting in a delay in tumors reaching the experimental endpoint (Fig. 3i). Similar tumor growth suppression was also observed in 22RV1 xenografts with *IGF1R-AS1* knockdown (Supplementary Fig. 3k–n). These results establish that *IGF1R-AS1* is required for the growth of *IGF1R-AS1*-positive prostate and lung cancer cells in vitro and in vivo.

To dissect how *IGF1R-AS1* contributes to cancer progression, we conducted RNA-seq in VCaP cells with *IGF1R-AS1* knockdown and compared the results to control cells. Gene Set Enrichment Analysis (GSEA) revealed that genes downregulated by *IGF1R-AS1* knockdown

**Fig. 4 | The *IGF1R-AS1* protein interactome reveals its interactions with chromatin remodeling complexes. a** FISH analysis of *IGF1R-AS1* in VCaP cells. (Left) *IGF1R-AS1* FISH signal (magenta) in wild-type (WT) cells (top) and after siRNA knockdown (bottom). DAPI stains nuclei (blue). Scale bar: 5 μm. (Right) Quantification of *IGF1R-AS1* foci distribution in WT cells (*n* = 4 biological replicates). Unpaired Student's *t*-test, two-sided. *p*\*\*\*=0.0009. **b** Workflow for detecting the *IGF1R-AS1* interactome. **c** High-confidence interactions between *IGF1R-AS1* and chromatin remodeling complex subfamily members are highlighted, with spectral counts indicating interaction strength. Additional interacting proteins are listed in Supplementary Data 2. **d** Binding of *IGF1R-AS1* to SMARCA1 and SMARCA4 was detected in *IGF1R-AS1* RNA pulldown assays. *IGF1R* sense RNA (*IGF1R-S1*) and streptavidin beads alone served as negative controls. **e** RIP-qPCR analysis of *IGF1R-AS1* enriched by anti-SMARCA1 and anti-SMARCA4 antibodies in VCaP cells (*n* = 3 biological replicates). *SChLAP1*, a known SWI/SNF-interacting lncRNA, was included as a positive control. Unpaired Student's *t*-test, two-sided. *p*\*\*\*\*<0.0001. **f, g** Western blotting analysis of recombinant SMARCA4 proteins and SMARCA1 proteins pulled down by *IGF1R-AS1* RNA in vitro binding assay. *IGF1R-S1* and streptavidin beads served as negative controls. **h** The secondary structure model of

*IGF1R-AS1* is predicted by RNAfold webserver based on minimum free energy algorithm. **i** Western blotting analysis of SMARCA1 and SMARCA4 pulled down by full-length and truncated *IGF1R-AS1*. **j, k** Schematic representation of Flag-tagged full-length SMARCA4 (**j**) and SMARCA1 (**k**), and its truncations, and RIP-qPCR in VCaP cells showing *IGF1R-AS1* enrichment by Flag-tagged full-length SMARCA4 (**j**) and SMARCA1 (**k**), and its truncations (*n* = 3 biological replicates). **l, m** GSEA analysis of RNA-seq data from *SMARCA4* knockdown (**l**) and *SMARCA1* knockdown (**m**) in VCaP cells. **n–q** qPCR analysis (**n, o**, *n* = 3 biological replicates) and western blotting analysis (**p, q**) of *MYC* mRNA and protein levels following *SMARCA4* or *SMARCA1* depletion in VCaP cells. Unpaired Student's *t*-test in (**n** and **o**), two-sided. *p*\*\*\*\*<0.0001. **r, s** VCaP cell proliferation curves following *SMARCA4* (**r**, *n* = 3 biological replicates) or *SMARCA1* (**s**, *n* = 3 biological replicates) depletion. Unpaired Student's *t*-test, two-sided. *p*\*\*\*\* (**r**, siNC vs siSMARCA4-1#, day 5) < 0.0001, *p*\*\*\* (**r**, siNC vs siSMARCA4-2#, day 5) = 0.0002, *p*\*\*\* (**s**, siNC vs siSMARCA1-1#, day 5) = 0.0002, *p*\*\*\* (**s**, siNC vs siSMARCA1-2#, day 5) = 0.0004. Data in (**a, e, j, k, n, o, r**, and **s**) are shown as mean ± SEM. Experiments in (**d, f, g, i, p**, and **q**) were biologically repeated three times. Source data of this Figure are provided as Source data file.

---

were enriched in cell-cycle-related biological processes, including "G2M checkpoint", "MYC targets", and "E2F targets" (Fig. 3j). We further validated these GSEA findings by implementing a previously-reported TF activity analysis[17], which nominated MYC as the primary dysregulated TF in *IGF1R-AS1* knockdown cells (Fig. 3k). Additionally, we observed significant enrichment of MYC targets among *IGF1R-AS1*-regulated genes in prostate cancer tissue samples (Supplementary Fig. 3o). Consistent with RNA-seq analysis, both the transcript and protein levels of MYC decreased after *IGF1R-AS1* knockdown in VCaP, 22RV1, and H727 cells, while the expression of *IGF1R* and *IGF1R* nearby genes *SYNM* and *TTC23* were not dramatically affected (Fig. 3l, m and Supplementary Fig. 3p–s). Reintroducing *IGF1R-AS1* in *IGF1R-AS1*-knockdown VCaP cells rescued the expression of *MYC* (Supplementary Fig. 3t, u), and overexpressing *IGF1R-AS1* in LNCaP cells also increased the expression of *MYC* (Supplementary Fig. 3v, w). Additionally, MYC-positive targets decreased significantly in *IGF1R-AS1* knockdown cells, while MYC-negative targets increased (Supplementary Fig. 3x). These findings highlight a non-canonical role for lncRNA *IGF1R-AS1*, which does not regulate *IGF1R* in *cis*, but functions in *trans* to regulate *MYC*.

## *IGF1R-AS1* interactome analysis reveals its interaction with chromatin remodelers

To explore the molecular mechanisms underlying the oncogenic role of *IGF1R-AS1*, we began by assessing its subcellular localization using fluorescence in situ hybridization (FISH) with custom-designed probes targeting *IGF1R-AS1* in VCaP cells (Fig. 4a). The FISH analysis revealed that *IGF1R-AS1* is primarily localized in the nucleus (Fig. 4a), suggesting its potential involvement in nuclear processes such as chromatin regulation. To confirm the probe's specificity, we performed FISH after siRNA-mediated *IGF1R-AS1* silencing, which showed an almost complete absence of nuclear signal, validating the initial observation (Fig. 4a and Supplementary Fig. 4a, b).

We next performed an RNA pull-down assay followed by mass spectrometry to define the *IGF1R-AS1* interactome in VCaP cells (Fig. 4b). After excluding low-confidence candidates, we identified 302 proteins associated with biotin-labeled *IGF1R-AS1* (Supplementary Data 2). Functional annotation of the *IGF1R-AS1* interactome revealed the enrichment of ATP-dependent chromatin remodelers (Supplementary Fig. 4c), involving distinct chromatin remodeling complex subfamily members (Fig. 4c). Subsequent immunoblot analyses of the lncRNA-bound proteins (Fig. 4d and Supplementary Fig. 4d) and antibody-based RNA immunoprecipitation followed by qPCR (Fig. 4e and Supplementary Fig. 4e, f) validated the interactions between *IGF1R-AS1* and both SMARCA4 and SMARCA1, representing the SWI/SNF and ISWI chromatin remodeling complexes, respectively. To determine whether these interactions are direct, we performed in vitro

binding assays using biotin-labeled *IGF1R-AS1* and recombinant His-tagged SMARCA4 or Flag-tagged SMARCA1 proteins, confirming direct binding (Fig. 4f, g). To map *IGF1R-AS1* regions mediating these interactions, we predicted its secondary structure (Fig. 4h) and conducted RNA pull-down assays with truncated *IGF1R-AS1* fragments (Fig. 4i). This analysis revealed that nucleotides 512–645 bind SMARCA4, whereas nucleotides 177–407 and 408–511 interact with SMARCA1 (Fig. 4i). Both SMARCA4 and SMARCA1 contain an ATPase domain within their central region. To identify the protein domains responsible for *IGF1R-AS1* binding, we cloned the N-terminal, ATPase, and C-terminal domains of SMARCA4 and SMARCA1 into Flag-tagged lentiviral constructs, expressed them in VCaP cells, and performed RIP using anti-Flag antibody followed by qPCR (Fig. 4j, k and Supplementary Fig. 4g, h). *IGF1R-AS1* was predominantly associated with the ATPase domain of SMARCA4 (Fig. 4j) and the C-terminal domain of SMARCA1 (Fig. 4k).

To examine whether *IGF1R-AS1* regulates the MYC pathway through its interactions with the SWI/SNF and ISWI complexes, we conducted RNA sequencing in VCaP cells with *SMARCA4* or *SMARCA1* knockdown, alongside control cells. GSEA analysis revealed that MYC targets were among the top-ranked hallmark gene signatures associated with SMARCA4 and SMARCA1 (Fig. 4l, m). We further validated these findings by showing that depletion of either *SMARCA4* or *SMARCA1* significantly reduced *MYC* expression at both the transcriptional and protein levels in VCaP, H727, and 22RV1 cells (Fig. 4n–q and Supplementary Fig. 4i–l). Additionally, knockdown of *SMARCA4* or *SMARCA1* led to a reduction in cell proliferation of VCaP, H727, and 22RV1 cells (Fig. 4r, s and Supplementary Fig. 4m, n), consistent with previous reports[16,18]. These findings suggest that the modulation of MYC signaling by *IGF1R-AS1* involves its interactions with chromatin remodelers.

## *IGF1R-AS1* expression induces extensive chromatin changes, with a predominant impact on the *MYC* promoter-enhancer interaction network

As *IGF1R-AS1* interacts with chromatin remodelers, we hypothesized that it influences chromatin accessibility, potentially leading to aberrant *MYC* expression. To investigate this, we conducted ATAC-seq experiments in VCaP cells transfected with *IGF1R-AS1* siRNA or control siRNA. We identified 12,808 sites with significantly increased accessibility and 9244 sites with decreased accessibility due to siRNA-mediated *IGF1R-AS1* depletion (Fig. 5a). To map which genomic regions are co-regulated by *IGF1R-AS1* and chromatin remodelers, we leveraged publicly available SMARCA4 ChIP-seq data in VCaP cells that enabled genome-wide identification of direct SWI/SNF binding sites (Fig. 5b). By comparing ATAC-seq peak density at these SMARCA4

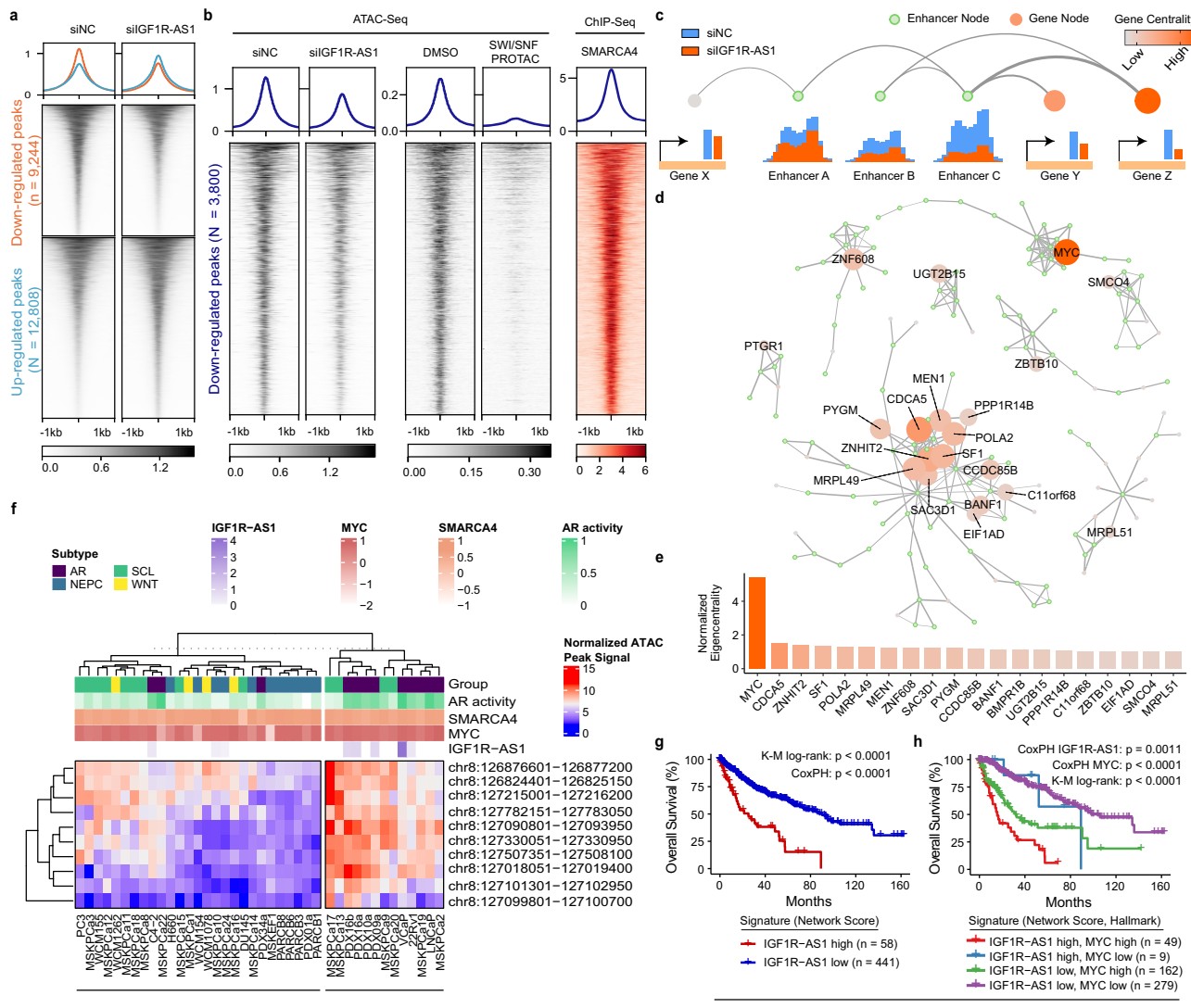

**Fig. 5 | *IGF1R-AS1* regulates enhancer activities, impacting *MYC* enhancer-promoter interaction network. a** Genome-wide chromatin accessibility changes following *IGF1R-AS1* depletion. Heatmap shows ATAC-seq signal intensity (±1 kb from peak center) for 9244 regions with decreased accessibility (top) and 12808 regions with increased accessibility (bottom). **b** Overlap of *IGF1R-AS1* depletion-induced accessibility changes with SMARCA4 binding sites. 3800 regions with reduced accessibility coincide with SMARCA4 ChIP-seq peaks (right). These peaks are completely lost upon treatment with the SWI/SNF PROTAC degrader AU-15330. **c** Schematic for constructing an enhancer-promoter (E-P) interaction network. **d** Top eight E-P interaction subnetworks by node size highlight key regulatory hubs like *MYC*, with node size and color intensity reflecting centrality within each network. **e** *IGF1R-AS1* predominantly influences the E-P subnetwork of *MYC*, as indicated by the *MYC* gene's highest normalized centrality score. **f** K-Means clustering

of 40 prostate cancer models, including mCRPC organoids, xenografts, and cell lines, based on ATAC-seq signals from the top 10 MYC-connected enhancers. *IGF1R-AS1* expression was mainly found in the high *MYC* enhancer activity group. **g** Kaplan–Meier (K–M) survival analysis of mCRPC patients (*n* = 499) stratified by *IGF1R-AS1* network activity score at optimal cutpoints. Cox proportional-hazards regression model (CoxPH) *p*-value indicates significance of *IGF1R-AS1* network activity as a predictor in a penalized spline Cox proportional hazards model. **h** Kaplan–Meier survival analysis of patients (*n* = 499) stratified by both *IGF1R-AS1* network activity and MYC pathway activity levels demonstrates poorest survival in *IGF1R-AS1*-high/*MYC*-high group. *p*-value indicates significance of *IGF1R-AS1* network activity as a predictor in a penalized spline Cox proportional hazards model after stratification by *MYC* expression.

binding sites, we distinguished between accessible regions directly influenced by *IGF1R-AS1*'s chromatin remodeling activity and those driven by other factors. Notably, 41.1% of the ATAC-seq peaks that were decreased by *IGF1R-AS1* knockdown (3800 sites) display a significant overlap with SMARCA4 binding sites (*p* < 0.0001, hypergeometric test). These decreased peaks exhibited the strongest association with SMARCA4 binding, surpassing both the increased and retained sites (Supplementary Fig. 5a). Additionally, we observed a widespread loss of ATAC-seq signals at the 3800 genomic sites in VCaP cells that displayed a decrease in peak signal following treatment with a SWI/SNF PROTAC[16] (Fig. 5b). Genomic annotation further revealed that over 86% of these sites are enhancers (distal intergenic and intronic regions)

(Supplementary Fig. 5b), suggesting a role for *IGF1R-AS1* in regulating SWI/SNF-dependent enhancer accessibility.

To identify genes regulated by *IGF1R-AS1*-associated enhancers, we developed an algorithm for constructing a gene-peak regulatory network. This approach integrated differentially accessible peaks from ATAC-seq data, differentially expressed genes from RNA-seq data, and enhancer-promoter interactions from high-throughput chromatin contact mapping (Fig. 5c, See "Methods" for details). Our analysis revealed 1158 gene-peak connections involving 674 unique peaks and 665 genes genome-wide (Supplementary Fig. 5c). *IGF1R-AS1* appeared to regulate gene-peak subnetworks characterized by multiple enhancer-promoter connections, forming promoter-enhancer hubs[19].

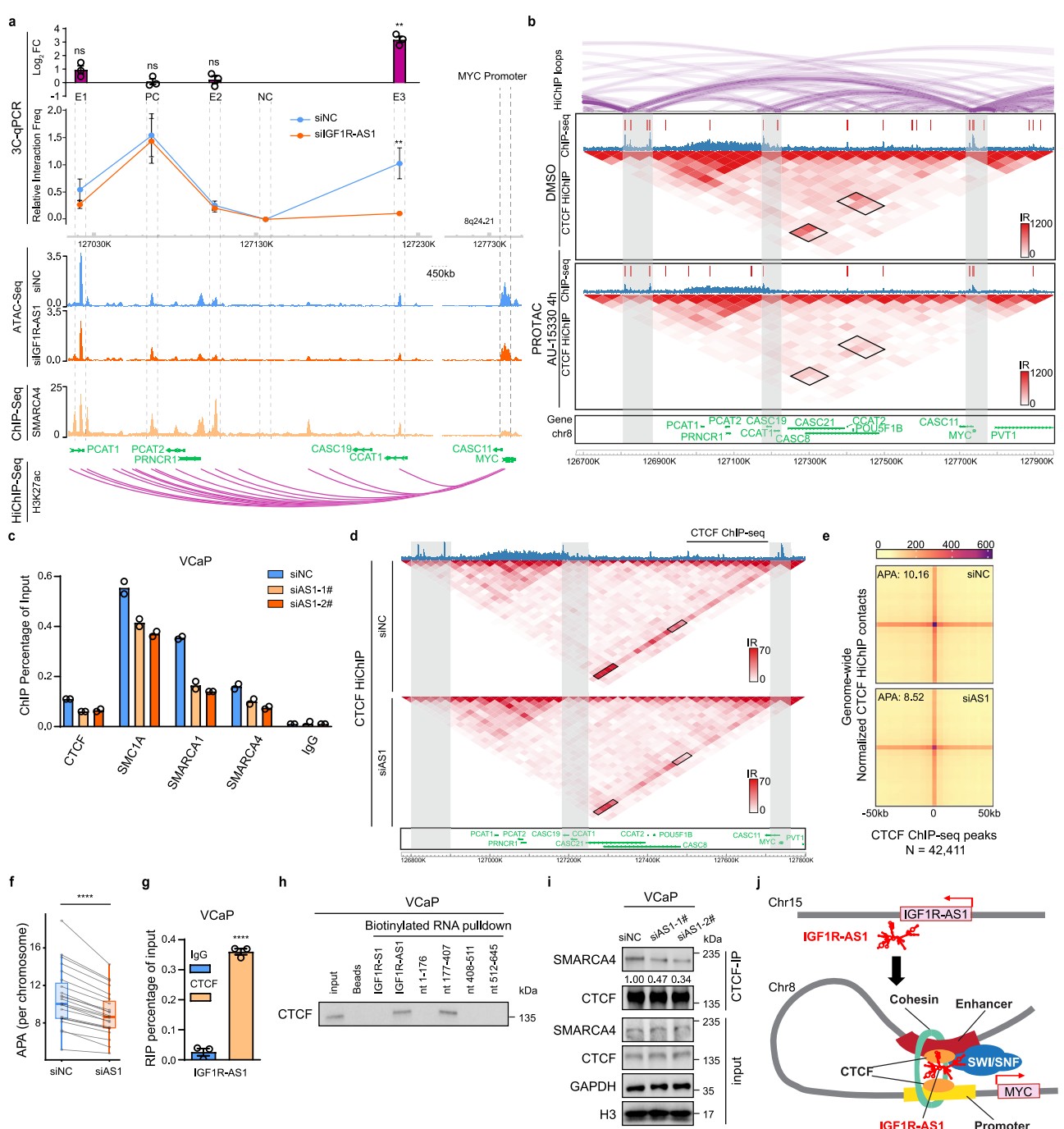

Of these, 176 genes (26.5%) were linked to five or more peaks (Supplementary Fig. 5d). To identify key promoter-enhancer hubs mediated by *IGF1R-AS1*, we assessed each network node's influence using eigencentrality analysis (See "Methods" for details). A high eigencentrality score indicates that a node has dense connections to other highly scoring nodes, signifying a hub of high transcriptional activity. *MYC* emerged as the highest-scoring gene, residing within a subnetwork containing 19 distinct enhancer nodes (Fig. 5d, e). Notably, 17 of these *MYC*-connected enhancer nodes ranked among the top 20 for eigencentrality across all enhancer nodes (Supplementary Fig. 5e). Similar network analysis of SMARCA4, the *IGF1R-AS1* interacting partner, also identified *MYC* as the top-ranked gene (Supplementary Fig. 5f), reinforcing *MYC*'s position as the primary promoter-enhancer hub regulated by *IGF1R-AS1*. Additionally, network analysis of SMARCA1 knockdown ATAC-seq similarly identified *MYC* among the top-ranked genes (Supplementary Fig. 5g), consistent with both SWI/

SNF and ISWI complexes contributing to *IGF1R-AS1*-mediated regulation of *MYC*.

To validate *IGF1R-AS1*'s regulation of *MYC* expression and enhancer activities across multiple cancer models, we analyzed a publicly available dataset containing paired ATAC-seq and RNA-seq data from 40 samples, including 22 mCRPC organoids, 6 patient-derived xenografts, and 12 derived or traditional prostate cancer cell lines[20]. Focusing on the top 10 *MYC*-associated enhancers identified in the *IGF1R-AS1* network, we classified samples into low and high *MYC* enhancer activity groups using K-Means clustering of ATAC-seq signals (Fig. 5f). Notably, *IGF1R-AS1*-positive samples predominantly clustered in the high *MYC* enhancer activity group (5/13, 39%, vs. 3/27, 11%), including VCaP cells. *IGF1R-AS1* expression was significantly elevated in this group (Supplementary Fig. 5h). Consistently, *IGF1R-AS1*-positive samples displayed increased *MYC* enhancer accessibility and expression (Supplementary Fig. 5i, j), aligning with their higher

**Fig. 6 | *IGF1R-AS1* modulates CTCF-mediated chromatin loops. a** Multi-panel visualization of the enhancer regions upstream of *MYC*: (top) Log fold change of interaction frequencies (siNC/siAS1) at different regulatory elements, showing significant change at E3, *n* = 3 biological replicates; (middle) 3C-qPCR analysis demonstrating loss of chromatin looping between *MYC* promoter and the E3 enhancer locus upon *IGF1R-AS1* depletion, *n* = 3 biological replicates; (bottom) genome browser tracks displaying ATAC-seq profiles in control and *IGF1R-AS1*-depleted cells, SMARCA4 binding by ChIP-seq, and H3K27ac HiChIP interactions connecting the *MYC* promoter to upstream enhancers. Unpaired Student's *t*-test, two-sided. *p*****=0.0049. ns not significant. **b** CTCF HiChIP-seq heat maps within the *MYC* gene locus in VCaP cells treated with DMSO or the SWI/SNF degrader AU-15330 (1 μM) for 4 h (bin size = 50 kb). ChIP-seq read-density tracks for CTCF are overlaid for both conditions. HiChIP loops indicate read-supported *cis* interactions within the locus. Grey highlights mark key regulatory regions, including enhancers and the *MYC* promoter. IR interaction reads. Gene annotations are shown below the heat maps, highlighting the *MYC* locus and adjacent lncRNA genes. **c** ChIP-qPCR shows decreased binding of CTCF, cohesin (SMC1A), and chromatin remodelers (SMARCA1/A4) at the E3 enhancer after *IGF1R-AS1* knockdown. *n* = 2 biological replicates. **d** CTCF HiChIP-seq heat maps within the *MYC* gene locus in VCaP cells in control and after *IGF1R-AS1* knockdown. (bin size = 25 kb). Overlaid is ChIP-seq read-density track for CTCF in DMSO from (**b**). Grey highlights mark key regulatory regions. IR interaction reads. **e** Genome-wide aggregate peak analysis (APA) plots in CTCF HiChIP using square root VC normalization. APA values indicate ratio between signal at CTCF ChIP-seq peaks (middle pixel) and mean signal at bottom left 3 × 3 pixels. **f** APA values for each chromosome (chr1-22, chrX) in control and *IGF1R-AS1* knockdown cells, showing that CTCF signal enrichment goes down in *IGF1R-AS1* knockdown cells across all chromosomes (*n* = 23, chr1-22, chrX). Paired Wilcoxon signed-rank test, *p*****<0.0001. Boxplots are shown as the median (center line), interquartile range (box bounds, 25th to 75th percentile), and whiskers extending to 1.5× the interquartile range, and points beyond the whiskers represent outliers. **g** RIP-qPCR analysis of *IGF1R-AS1* enrichment by anti-CTCF antibody in VCaP cells (*n* = 3 biological replicates). Unpaired Student's *t*-test, two-sided. *p*****<0.0001. **h** Western blotting analysis of CTCF pulled down by full-length and truncated *IGF1R-AS1*. **i** Co-immunoprecipitation (co-IP) of SMARCA4 with CTCF in VCaP cells upon *IGF1R-AS1* depletion. **j** A model demonstrating *IGF1R-AS1* promotes *MYC* transcription by mediating SWI/SNF-CTCF interaction and facilitating chromatin loops between *MYC* enhancer and promoter. Data in (**a** and **g**) are shown as mean ± SEM. Experiments in (**h** and **i**) were biologically repeated three times. Source data of this Figure are provided as Source data file.

representation in the high *MYC* enhancer activity group. Reconstructing the *IGF1R-AS1* regulatory network using these samples' ATAC-seq and RNA-seq data confirmed *MYC* as the primary hub gene (Supplementary Fig. 5k), consistent with findings in VCaP cells.

To evaluate the clinical relevance of *IGF1R-AS1*-regulated *MYC* transcription, we analyzed RNA-seq data from 499 mCRPC tumor biopsies profiled using the Tempus xR assay (Tempus AI, Inc., Chicago, IL) (https://www.tempus.com/wp-content/uploads/2023/06/Tempus-xR_Validation.pdf)[21]. As this assay employs a hybrid capture approach to profile the transcriptome, it cannot directly detect *IGF1R-AS1* expression due to the lncRNA's unannotated status. To address this, we developed an *IGF1R-AS1* activity score based on the gene signature derived from its regulatory network (See "Methods" for details). High *IGF1R-AS1* activity was significantly associated with poor overall survival (*p* < 0.0001; Fig. 5g). Stratifying patients by both *IGF1R-AS1* activity and MYC hallmark gene signature revealed that individuals with high levels of both had the worst clinical outcomes, with *IGF1R-AS1* activity remaining a significant predictor of survival in this stratified analysis (*p* = 0.0011; Fig. 5h). Independent validation in a microarray based primary prostate cancer cohort[22] and SU2C mCRPC cohort[23] confirmed that high *IGF1R-AS1* network activity predicts worse disease-free and overall survival, respectively (Supplementary Fig. 5l, m). Additionally, *IGF1R-AS1* further stratified outcomes among *MYC*-high patients and showed superior prognostic performance compared to Prolaris cell-cycle progression markers[24-27] (Supplementary Fig. 5m).

A prior study indicates that concurrent high MYC and low AR transcriptional programs promote prostate cancer progression toward metastatic disease[28]. Analysis of our Tempus mCRPC cohort confirmed these findings, with MYC-high/AR-low patients showing significantly worse survival (*p* < 0.0001; Supplementary Fig. 5n). Notably, *IGF1R-AS1* activity was elevated in this aggressive subgroup (Supplementary Fig. 5o), suggesting that the *IGF1R-AS1*-mediated regulation of *MYC* transcription may contribute to driving this molecular subtype in mCRPC. Taken together, these findings suggest that *IGF1R-AS1* amplifies the aggressiveness of MYC-driven tumors, identifying a subset of particularly high-risk patients.

### *IGF1R-AS1* orchestrates CTCF-anchored chromatin loops connecting *MYC* promoter with distal *cis*-regulatory enhancer elements

Given the role of *IGF1R-AS1* in regulating *MYC* enhancer activities, we hypothesized that it modulates chromatin looping between *MYC* enhancers and its promoter. Our data showed that architectural proteins CTCF and cohesin subunit SMC1A were part of the *IGF1R-AS1*

interactome (Supplementary Data 2), indicating a potential link between CTCF/cohesin-mediated looping and *IGF1R-AS1*. To explore this, we conducted quantitative chromosome conformation capture (3C-qPCR) in *IGF1R-AS1* knockdown and control VCaP cells to assess interaction frequencies between the *MYC* promoter and its enhancers. We chose two *IGF1R-AS1*-regulated *MYC* enhancers spanning the *PCAT1-PRNCR1* loci (E1 and E2), which are known prostate cancer-associated *cis*-regulatory elements (CREs)[29,30]. We tested another enhancer region at the *CCAT1* locus (E3), which is among the top five affected *MYC* enhancers along with E1 and E2 predicted by our network analysis (Supplementary Fig. 5e). We also included a negative control (NC) region lacking evidence of interaction with *MYC* promoter and a positive control (PC) region with a previously validated interaction with *MYC* promoter[31]. Quantitative 3C analysis revealed that *IGF1R-AS1* depletion had differential effects on *MYC* promoter-enhancer interactions: while maintaining most interactions with PC, E1, and E2 regions, it caused a complete loss of interaction with the E3 enhancer region (Fig. 6a). These findings suggest that *IGF1R-AS1* selectively influences long-range chromatin interactions within the *MYC* regulatory network, with particularly critical impact on the loop connecting the *MYC* promoter with the E3 enhancer in the *CCAT1* locus.

A previous study in colon cancer demonstrated that *CCAT1-L* lncRNA, a *CCAT1* isoform with longer extra 3′ extension, can interact with CTCF to enhance CTCF-mediated looping between the *CCAT1* locus, where E3 is located, and the *MYC* locus[4]. Another study also reported that *CCAT1-5L*, a *CCAT1* isoform with an extra 5′ extension, could regulate *MYC* transcription in HeLa cells via interaction with *MYC* promoter RNAs and eRNA *PVT1* to modulate long-range chromatin looping[32]. However, *CCAT1* and its long isoforms are not expressed in prostate cancer cell lines including VCaP cells (Supplementary Fig. 6a) and *CCAT1* exhibits very low expression levels across prostate cancer tissue samples compared to other cancers (Supplementary Fig. 6b). Despite this, E3 accessibility remains correlated with *MYC* expression across TCGA pan-cancer cohorts, including prostate cancer, as demonstrated by the correlation analysis of publicly available ATAC-seq and RNA-seq data from TCGA samples[33] (Supplementary Fig. 6c). Furthermore, E3 accessibility is significantly elevated in *IGF1R-AS1* positive prostate cancer samples (Supplementary Fig. 6d). These findings suggest that the chromatin loop between E3 and the *MYC* locus in prostate cancer is likely *trans*-regulated by *IGF1R-AS1*, rather than driven by *cis*-regulation through *CCAT1* or its isoforms. To assess whether this E3-*MYC* chromatin loop is CTCF-mediated, we analyzed publicly available ChIP-seq data for CTCF and the cohesin subunit RAD21. Our analysis revealed that both CTCF and cohesin bind to the

E3 region across different cell types, with CTCF showing the strongest binding in VCaP cells (Supplementary Fig. 6e). Furthermore, CTCF HiChIP data from VCaP cells confirmed strong interactions between E3 and the *MYC* promoter (Fig. 6b), providing direct evidence that this long-range looping is CTCF-mediated.

To further investigate how *IGF1R-AS1* impacts chromatin loops, we conducted ATAC-seq footprinting analysis for TFs using binding profiles annotated by the JASPAR database[34]. Aggregate footprint analysis showed that *IGF1R-AS1* depletion globally reduced CTCF binding at open chromatin regions co-regulated by *IGF1R-AS1* and SMARCA4 (Supplementary Fig. 6f). Given that chromatin remodelers regulate CTCF-mediated 3D architecture[35,36] and *trans*-acting lncRNAs can direct the SWI/SNF complex to specific enhancers[37], we hypothesized that *IGF1R-AS1* enhances the recruitment of chromatin remodelers and CTCF to *MYC* enhancers, thereby facilitating CTCF/cohesin-mediated loops between *MYC* enhancers and promoters. Supporting this, analysis of CTCF ChIP-seq and HiChIP data from SWI/SNF PROTAC-treated VCaP cells showed that SWI/SNF inactivation notably disrupted CTCF binding and reduced CTCF-mediated chromatin loops connecting *MYC* promoter with distal CREs, including the E3 enhancer region (Fig. 6b). Consistently, *IGF1R-AS1* knockdown reduced the binding of SMARCA4, SMARCA1, CTCF, and the cohesin subunit SMC1A at the E3 enhancer (Fig. 6c), without altering their protein abundance (Supplementary Fig. 6g). To more comprehensively evaluate CTCF-mediated chromatin interactions, we performed CTCF HiChIP in control and *IGF1R-AS1*−depleted VCaP cells. CTCF HiChIP confirmed that *IGF1R-AS1* depletion disrupted *MYC* promoter−upstream enhancer loops (Fig. 6d). Genome-wide aggregate peak analysis (APA) demonstrated a significant reduction in CTCF loop strength following *IGF1R-AS1* knockdown (Fig. 6e), accompanied by decreased CTCF signal enrichment across all chromosomes (Fig. 6f). Previous investigations demonstrated that CTCF and SMARCA4 physically interact and colocalize to enhancers and regulate gene expression[35,38,39]. Our *IGF1R-AS1* interactome analysis indicates that CTCF binds *IGF1R-AS1* (Supplementary Data 2). We validated this interaction through CTCF RIP-qPCR (Fig. 6g) and *IGF1R-AS1* RNA pulldown, which further showed that *IGF1R-AS1* interacts with CTCF through nucleotides 177−407 (Fig. 6h), whereas it interacts with SMARCA4 through nucleotides 512−645 (Fig. 4h). Furthermore, CTCF co-IP demonstrated that *IGF1R-AS1* knockdown reduced SMARCA4−CTCF interaction (Fig. 6i), indicating that *IGF1R-AS1* acts as a scaffold that stabilizes this interaction. Collectively, these findings reveal a mechanism by which the *trans*-acting lncRNA *IGF1R-AS1* promotes long-range chromatin looping to activate oncogenic *MYC* transcription by acting as a scaffold to increase the interaction between SMARCA4 and CTCF (Fig. 6j).

## Discussion

Non-coding RNAs were once dismissed as "transcriptional noise." However, mounting evidence has highlighted their biological functions and clinical significance across diverse diseases[40]. Advances in high-throughput technologies and bioinformatics have fueled interest in lncRNAs[41], which exhibit tissue-specific expression patterns and play crucial roles in complex physiological and pathological processes. Efforts to identify lncRNAs have revealed numerous regulatory candidates in human diseases, with several emerging as key regulators in carcinogenesis[42]. In our study, we analyzed transcriptomes from mCRPC and identified prostate cancer-associated lncRNAs. Among these, *IGF1R-AS1* stands out as a lncRNA regulated by a SWI/SNF-associated SE. *IGF1R-AS1* exerts an oncogenic role by acting as a scaffold and interacting with chromatin remodelers and architectural proteins, modulating chromatin loops between the *MYC* promoter and its distal CREs, leading to upregulation of the MYC signaling pathway.

In this study, we investigated *IGF1R-AS1*, a top-ranked SE-associated unannotated lncRNA, and validated its oncogenic function. Given that SEs play a pivotal role in tumorigenesis and cancer progression through robust transcriptional activation of oncogenes[43,44], other SE-associated lncRNAs identified in our analysis may similarly contribute to oncogenic processes in cancer cells. However, the functional mechanisms of these unannotated SE-associated lncRNAs, whether they regulate transcription through interactions with chromatin factors or engage in post-transcriptional regulatory mechanisms, remain largely unexplored. To comprehensively understand the functions of SE-associated lncRNAs in transcriptional and chromatin regulation, particularly those acting in *trans* rather than *cis*, high-throughput approaches are essential for screening genome-wide interactions with molecular components such as chromatin regulators and DNA regulatory elements. Emerging techniques like GRID-seq[45], which enables the identification of global RNA-chromatin interactions, offer promising methodological solutions to address these research challenges.

MYC overexpression is a hallmark of various cancers and can be driven by multiple mechanisms, including genetic alterations, transcriptional regulation, and protein stabilization[46]. Recently, enhancer activation, particularly through SEs, has been identified as a key mechanism underlying MYC activation in cancer[30]. Previous studies have shown that *cis*-regulatory lncRNAs transcribed from SEs surrounding *MYC* can modulate long-range interactions between the *MYC* promoter and these SEs, as exemplified by *CCAT1* isoforms[4,32]. Another example is the *MYC* downstream lncRNA *PVT1*, whose promoter limits long-range interactions between the *MYC* promoter and intragenic enhancers within *PVT1*[47]. Our study demonstrates that *MYC* enhancer-promoter interactions can be regulated by a lncRNA acting in *trans*. This form of regulation arises from the intricate interplay among the lncRNA, chromatin remodelers, and architectural proteins. Due to the SWI/SNF complex's critical role in PCa biology and SMARCA4's alignment with *IGF1R-AS1* in the regulation of chromatin accessibility, we focused on investigating SMARCA4's participation in *IGF1R-AS1*-regulated *MYC* promoter-enhancer interactions. Notably, a recent study revealed that *trans*-acting lncRNAs guide the SWI/SNF complex binding to cell type-specific enhancers[37]. Based on the evidence presented in this study, it is conceivable that additional *trans*-acting lncRNAs, beyond *IGF1R-AS1*, may play critical roles in modulating *MYC* enhancer activities in a variety of human cancers. To scrutinize this hypothesis, integrative multi-omics approaches, encompassing RNA expression profiles, chromatin accessibility, and 3D chromatin looping, are essential for gaining a comprehensive understanding of the complex genomic regulations governing *MYC* expression.

Beyond the specific context of *MYC* regulation, our findings highlight the urgent need to define the broader landscape of tumor-specific lncRNAs across cancer types. Traditional omics approaches based on differential expression between healthy and cancerous tissues often miss lncRNAs uniquely expressed in tumors, particularly those with low abundance or high context-specificity[10]. As a result, computational methods are needed to detect these tumor-specific transcripts independently of their expression in normal tissue or reference annotations. Incorporating multi-omic data, including chromatin state, somatic alterations, and epigenetic modifications, not only improves candidate prioritization but also enhances sensitivity in detecting tumor-specific lncRNAs, as demonstrated by *IGF1R-AS1* in our study, which was nominated through integrated SE analysis. While these omics approaches provide valuable indirect evidence of functionality, rigorous experimental validation remains essential. High-throughput perturbation tools such as CRISPR-Cas13 and antisense oligonucleotides now make it feasible to systematically assess the functional relevance of tumor-specific lncRNAs in relevant cancer models and accelerate the discovery of unexplored cancer drivers and potential therapeutic targets[48].

In summary, our study revealed a non-canonical function for *IGF1R-AS1*, an intronic antisense lncRNA, which operates in *trans*, diverging from previous studies that primarily associated this lncRNA

class with *cis*-regulatory roles affecting the expression of cognate sense or neighboring genes[3]. Our findings provide compelling evidence that *IGF1R-AS1* activates the MYC signaling pathway by *trans*-regulating long-range chromatin contacts at the *MYC* locus, thereby driving cancer progression. This work highlights the critical roles of lncRNAs in shaping the 3D genome and opens new avenues for exploring the regulatory functions of *trans*-acting lncRNAs in gene regulation. Collectively, our findings contribute to the growing body of evidence supporting the importance of tumor-specific lncRNAs as potential cancer drivers and therapeutic targets, emphasizing the need for continued investigations into this largely unexplored dimension of cancer biology.

## Methods

### Transcriptome assembly of the SU2C cohort prostate cancer patient samples

The quality of the polyA-enriched RNA-seq data of 101 metastatic prostate cancer samples in the SU2C cohort was initially assessed using FastQC (version 0.11.8), with all raw reads meeting the required quality standards. The RNA-seq reads were then aligned to the human reference genome (hg38) using HISAT2 (version 2.2.1)[49], with the strand parameter set to RF (option "rna-strandness"). Based on the aligned reads, the transcriptome was reconstructed using StringTie (version 1.3.6)[50], with reference annotations (option "-G") from GENCODE (v37) and the strand parameter (option "−rf") set to RF for each of the 101 samples. StringTie's merge mode was then used to integrate all 101 meta-assemblies into a final transcriptome.

### LncRNA identification

To identify a high-confidence set of lncRNAs not annotated by GEN-CODE, several stringent filters were applied: (1) Filtering Overlapping Transcripts: Transcripts that overlapped with low complexity regions or ENCODE blacklist regions were discarded. The low complexity regions were sourced from a public dataset available at Figshare[51] and ENCODE blacklist regions were obtained from ENCODE[52]; (2) Length and Strand Information: Transcripts shorter than 200 bp or lacking strand information were excluded from further analysis; (3) Expression Level: Transcript expression levels were calculated using StringTie. Transcripts with low expression levels (FPKM < 1.0) and those appearing in fewer than five samples were removed; This prevalence threshold was selected to focus on recurrently expressed lncRNAs across multiple patients while filtering sporadic detections likely representing technical noise. (4) Proximity Filtering: Single-exon transcripts proximal (within 2000 bp) to protein-coding genes or other noncoding RNAs on the same strand were filtered out to eliminate fragments of annotated RNA; (5) Overlap with Annotated Genes: Transcripts that overlapped with the exons of protein-coding genes or noncoding RNAs, including canonical noncoding RNAs and long lncRNAs annotated by GENCODE (v37), were removed. Canonical noncoding RNAs included rRNA, tRNA, miRNA, snRNA, snoRNA, misc_RNA, sRNA, scaRNA, vaultRNA, ribozyme, and mitochondrial tRNA and rRNA. Two biotypes defined as "long non-coding RNAs" by GEN-CODE included: "lncRNA" and "TEC". Some lncRNAs have been reclassified as protein-coding genes in recent GENCODE annotation versions. GFFcompare[53] was used to compare the assembled transcripts with the reference annotations; (6) Coding Potential Analysis: To verify that the candidate lncRNAs were non-coding, five computational methods were employed: CPAT[54], CPC2[55], lncADeep[56], CNIT[57], and txCdsPredict from UCSC. CPC2, lncADeep, and CNIT provided predicted labels (coding/noncoding) for the transcripts. CPAT used a coding probability cutoff of 0.364, while txCdsPredict used a score cutoff of 800 to determine coding potential. Only transcripts consistently predicted to be noncoding by all five tools were retained.

Identified lncRNA were classified into four categories: intergenic, overlapping, intronic, and exonic based on their relationship with protein-coding genes. The classification scheme was adopted from Derrien et al.[58]. PhastCons scores[59] obtained from the UCSC genome browser were used to evaluate the evolutionary conservation for protein-coding genes, annotated lncRNAs and unannotated lncRNAs.

### Differentially expressed lncRNA identification in mCRPC compared to normal prostate tissues

RNA sequencing data from 101 mCRPC samples in the SU2C cohort were processed described above. Aligned BAM files from the TCGA-PRAD cohort were retrieved from the Genomic Data Commons (GDC)[60] data portal. Gene-level read quantification was performed using FeatureCounts (version 2.0.3)[61] software, utilizing GENCODE (v37) gene annotations. In addition to reference annotations, lncRNAs identified in this study were incorporated into the analysis. Differential gene expression analysis was conducted using the edgeR[62] statistical package. LncRNAs were considered differentially expressed based on stringent criteria: false discovery rate (FDR) < 0.05 and absolute log2 fold change (|Log2FC|) > 1.5 when comparing mCRPC samples to normal prostate tissues.

### ChIP-seq and super-enhancer analysis

All ChIP-seq datasets were reprocessed by aligning the raw reads to the GRCh38 reference genome using BWA-MEM[63] (version 0.7.17) with default parameters. Normalized alignment coverage files were generated using the bamCoverage module from deepTools[64] (version 3.5.1) with parameters: −binSize 20, −normalizeUsing CPM, and −smoothLength 60. The deepTools modules computeMatrix and plotProfile were used to visualize ChIP signal at the center of target regions, including peak summits and transcription start sites (TSS).

For integration with *IGF1R-AS1* ATAC-seq analysis, SMARCA4 ChIP-seq data for VCaP was obtained from a previous study[15]. Raw FASTQ files were processed using the ENCODE ChIP-seq transcription factor pipeline (version 2.2.2). Reads were aligned with BWA-MEM, and peaks were called for each replicate using SPP[65]. Peak sets were analyzed and pooled for Irreproducible Discovery Rate (IDR) analysis, with optimal IDR peaks selected for downstream analyses.

Super-enhancers (SEs) were identified using the Rank Ordering of Super-Enhancers (ROSE) algorithm[12]. H3K27ac ChIP-seq peaks were initially used to define enhancers, applying the following criteria: peaks within ±3 kb of TSSs were excluded. Remaining H3K27ac peaks were designated as putative enhancers. Enhancers within ±12.5 kb of each other were stitched together, scored, and ranked based on H3K27ac ChIP-seq signal intensity. Enhancer rank was plotted against enhancer density, and regions above the inflection point of the curve were defined as SEs. ROSE algorithm default parameters were used to assign SEs and typical enhancers to genes. To visualize ChIP-seq signal at SE regions, averaged profiles were generated with ngsplot[66]. Genes were assigned to SEs using the ROSE program if their TSS was located within 50 kb of an SE.

### *IGF1R-AS1* 3′ rapid amplification of cDNA ends (3′-RACE)

RNA extracted from VCaP and H727 cells was subjected to 3′-RACE using a second-generation 5′/3′ RACE kit (Roche, Cat. No. 03353621001) according to the manufacturer's instructions. Briefly, 1 µg total RNA was used for first-strand cDNA synthesis using the oligo (dT)-anchor primer provided in the kit. An aliquot of the cDNA reaction was amplified by PCR using Quanta AccuStart II PCR SuperMix (Quanta Biosciences, 95137). The PCR utilized a forward primer anchored near the beginning of *IGF1R-AS1* exon 1 (5′-ACTATGTTTCAGCGGCTGGT-3′) and a reverse primer provided by the kit (5′-GACCACGCGTATC-GATGTCGAC-3′). The final 3′ RACE products were purified using a QIAquick PCR Purification Kit (Qiagen, 28104).

### PacBio iso-seq sequencing and data analysis

The amplified 3′-RACE products from VCaP and H727 cells were submitted to the University of Minnesota Genomics Center, where they

were converted into barcoded SMRTbell libraries using the Pacific Biosciences (PacBio) protocol for barcoded adapters for multiplex SMRT sequencing. The barcoded SMRTbell libraries were pooled and prepared for diffusion loading onto a PacBio Sequel instrument for sequencing. Raw data were processed using the CCS tool (version 6.0.0, https://ccs.how/) to generate circular consensus sequence (CCS) reads from the subreads generated during sequencing. LIMA (version 2.0.1, https://lima.how/) was then used for primer removal and demultiplexing, yielding full-length (FL) reads. ISOSEQ3 (version 3.4.0, https://isoseq.how/) was employed to refine these FL reads by removing polyA tails and artificial concatemers to produce full-length non-concatemer (FLNC) reads, which were further clustered into high-quality transcript isoforms. These isoforms were mapped to the GRCh38 human reference genome using minimap2[67] (version 2.17). Finally, we used SQANTI[68] to classify and quantify the full-length transcript isoforms.

## Expression analysis of *IGF1R-AS1* and *CCAT1* across cell lines and patient cohorts

The expression level of *IGF1R-AS1* was quantified using RNA-seq data from tumor and normal tissue samples across TCGA cohorts, accessed via the GDC data portal[60]. Briefly, samples with *IGF1R-AS1* characteristic junction (chr15: 98839896-98843675) were identified using RegTools (version 0.4.0)[69]. Next, for the samples with *IGF1R-AS1* characteristic junction, the Fragments Per Kilobase of transcript per Million mapped reads (FPKM) value of *IGF1R-AS1* were calculated using FeatureCounts[61] and in-house script based on the GTF annotation file concatenating GENCODE (v37) and *IGF1R-AS1* isoforms identified from PacBio Iso-Seq data. Pan-cancer expression levels of *CCAT1* were quantified using batch-corrected normalized expression values from UCSC Xena data platform[70].

*IGF1R-AS1* expression was further evaluated across prostate cancer cell lines (LNCaP, VCaP, 22RV1, PC3, DU145, NCI-H660, and MDA-PCa-2b) using RNA-Seq data obtained from the Cancer Cell Line Encyclopedia (CCLE)[71]. FPKM values were calculated using FeatureCounts (version 2.0.3), utilizing the *IGF1R-AS1* isoform annotations derived from PacBio Iso-Seq data. *CCAT1* expression in these cell lines was similarly quantified using the NCBI RefSeq annotation (GCF_000001405.40-RS_2024_08).

## RNA-seq data generation and downstream analysis

Total RNA was extracted using the RNeasy Plus Kit (Qiagen, 74134) and sequenced on an Illumina HiSeq 2500 instrument in High-output Mode to generate strand-specific paired-end reads. Sequencing quality and data integrity were inspected, and adaptor-trimmed reads were aligned to the human reference genome (GRCh38) using HISAT2. Gene-level read counts were quantified using FeatureCounts based on GENCODE annotations. Differential gene expression analysis was performed with edgeR[62], identifying differentially expressed genes between siRNA knockdown and control conditions at an adjusted *p*-value threshold of <0.05. Gene Set Enrichment Analysis (GSEA)[72] was conducted using log2 fold change values as the ranking metric to identify gene signatures associated with *IGF1R-AS1*, SMARCA1, and SMARCA4 in VCaP cells. Hallmark gene sets, oncogenic signatures, and curated gene sets were utilized for the analysis. Transcription factor activity following *IGF1R-AS1* knockdown was evaluated using a weighted mean approach as previously described[17], based on the CollecTRI transcription factor regulon database[73]. RNA-seq data of H727 cells treated with JQ1 and DMSO were processed following the same workflow as described above.

## ATAC-seq data generation and analysis

ATAC-seq was performed using ATAC-seq kit purchased from Active Motif. Briefly, 50,000 fresh cells with >90% viability were lysed to release nuclei. The nuclei pellet was treated with Tn5 transposase to simultaneously tag target DNA with sequencing adapters and fragment the DNA in a process termed "tagmentation". Following library amplification and purification, ATAC-seq libraries were subjected to high-throughput sequencing. Raw sequencing reads in FASTQ format were processed using the ENCODE ATAC-seq pipeline (version 2.2.1)[74]. Reads were first trimmed using Cutadapt (version 1.9.1)[75] and aligned to hg38 using Bowtie2 (version 2.2.6)[76]. Mapped reads overlapping blacklisted regions or mitochondrial DNA, as well as duplicate reads, were removed using SAMtools (version 1.7)[77] and PICARD MarkDuplicates (version 1.26) respectively. Peaks were called for all replicates using MACS2 (version 2.1.0)[78]. Final peak sets for each condition were generated using the optimal Irreproducible Discovery Rate (IDR) cut-off calculated by the ENCODE pipeline. For comparative analysis of publicly available ATAC-seq data of VCaP cells treated with DMSO and AU-15330, raw FASTQ files were obtained from the original study[16] and processed using the same pipeline described above.

Differential accessibility analysis between conditions was performed using the sliding windows approach implemented in the csaw R package (version 1.32)[79]. The genome was divided into 150 bp windows, with ENCODE hg38 blacklist regions excluded. ATAC-seq read counts were assigned to each window and normalized using a LOESS-based method[80] to account for compositional biases. Differential accessibility was determined using csaw's negative binomial generalized linear model, and significant windows were merged based on default *csaw* parameters. Peaks from the optimal IDR peak set generated by the ENCODE pipeline were classified as differentially accessible if they overlapped with a differentially accessible window meeting the significance criteria (FDR < 0.05 and |Log$_2$FC| > 0.25).

## Hi-C with chromatin immunoprecipitation (HiChIP)-seq data generation and analysis

HiChIP was performed following the published protocol[81] with minor modifications as described[82]. Briefly, chromatin from 5 million cells was crosslinked with 1% formaldehyde for 10 min, digested with the MboI restriction enzyme, end-repaired with dNTP including biotin-labeled dATP, and ligated with T4 DNA ligase. Ligated chromatin was sonicated with Qsonica Q800 (1.5 min total sonication time, 50% amplitude) and processed with ChIP using an CTCF antibody (Cell Signaling, #3418, 15 µL per HiChIP). ChIP-enriched DNA was de-crosslinked, purified, and processed with streptavidin beads to enrich ligated DNA fragments. Sequencing libraries were prepared using Illumina Tagment DNA Enzyme and Buffer Kit, and subjected to high-throughput sequencing. Raw sequencing reads in FASTQ format were processed using a custom NextFlow pipeline (https://github.com/ylab-hi/nf_hichip). Briefly, reads were aligned using BWA mem (version 0.7.19) with the –5SP option. Aligned paired reads were annotated with pairtools (version 1.1.3) using the parse function with –min-mapq 40, –walks-policy 5unique –max-inter-align-gap 30. Duplicate reads were removed with pairtools sort and partools dedup tools with basic options to obtain valid pairs files. Valid pairs were then converted to .hic format using juicer_tools (version 1.6).

For HiChIP contact heatmaps, .hic files were uploaded to the WashU Epigenome Browser (https://epigenomegateway.wustl.edu/) and screenshots from *MYC* loci were downloaded using default viewing conditions. For the aggregate peak analysis (APA) plots, .hic files were first converted to cool files at 5 kb resolution with VC_SQRT normalization using HiCExplorer (version 3.6) hicConvertMatrix. Aggregate contacts per chromosome for each replicate were then calculated using hicAggregateContacts function with parameters –range 50000:1000000, –numberOfBins 30, –operationType mean, –transform obs/exp. The resulting 31 × 31 matrix was averaged for over two replicates per condition (si*IGF1R-AS1* and siNC). APA scores were calculated by taking the ratio between the signal intensity at the center pixel divided by the mean intensity of a 3 × 3 square of pixels in the lower left corner. Finally, 21 × 21 matrixes were plotted in R using

ComplexHeatmap package (version 2.18) to highlight center enrichment.

## Gene-peak regulatory network construction

To construct the de novo gene-peak regulatory network, nodes representing either ATAC-seq peak regions or gene promoters were selected. ATAC-seq peak regions overlapping SMARCA4 ChIP-seq peaks and significantly downregulated upon *IGF1R-AS1* knockdown (FDR < 0.05, Log$_2$FC < −0.25) were included, alongside gene promoter regions located 2000 bp upstream or 200 bp downstream of transcription start sites (TSS). To ensure that ATAC-seq peaks represented enhancers, only those overlapping H3K27ac ChIP-seq peaks were retained, resulting in 3530 peaks (reduced from 3800). Gene promoter nodes were limited to those associated with genes significantly downregulated in *IGF1R-AS1* knockdown (FDR < 0.05, Log$_2$FC < −0.25).

Edges between nodes were established by integrating data from publicly available RNA Pol II ChIA-PET[83], H3K27ac HiChIP[16], and annotated prostate cancer-specific enhancer-gene links from EnhancerAtlas 2.0[84]. Connections were defined as follows:

1. **Peak-to-Gene Edges**: A peak node was connected to a gene node if there was a validated link between the peak region and the gene promoter in at least one data source.
2. **Peak-to-Peak Edges:** Two peak nodes were connected if there was evidence linking their corresponding regions.

Edge weights were calculated based on two criteria:

1. **Peak-to-Gene Edges**:
   - Ranked by the product of ATAC-seq and RNA-seq Log$_2$FC values in *IGF1R-AS1* knockdown (highest product ranked as 1).
   - Ranked by normalized ATAC-seq peak heights (highest value ranked as 1).
   - Final weights were computed as the product of these ranks and scaled between 0 and 1.

2. **Peak-to-Peak Edges**:
   - Ranked by the product of ATAC-seq Log$_2$FC values for both peaks in *IGF1R-AS1* knockdown.
   - Ranked by the mean normalized peak heights.
   - Final weights were computed as the product of these ranks and scaled between 0 and 1.

To assess the influence of each node, the weighted eigencentrality was calculated using the eigen_centrality() function from the igraph R package (v2.0.3). For a node *n*, the eigencentrality $C_n$ was computed as:

$$C_n = \frac{1}{\lambda} \sum_{i \in N} a_{n,i} ! C_i \qquad (1)$$

where *N* is the set of nodes connected to *n*, $a_{n,i}$ is the edge-weighted adjacency matrix, and λ is a scaling constant. To ensure the network was strongly connected, a "head node" was added with weak connections (0.01 weights) to all nodes, representing *IGF1R-AS1*'s regulatory influence. This adjustment minimally impacted eigencentrality calculations.

A parallel SMARCA4-based regulatory network was constructed using *SMARCA4* knockdown ATAC-seq and RNA-seq data to define nodes and edges. For networks derived from castration-resistant prostate cancer (CRPC) models, RNA Pol II ChIA-PET data from ENCODE prostate cancer cell lines (DU145, LNCaP, PC-3, VCaP) were used to define gene-peak links. *IGF1R-AS1*-regulated genes were identified as upregulated in *IGF1R-AS1*-positive samples (Wilcoxon rank-sum test, *p* < 0.05), and associated peaks were identified based on significant correlation between ATAC-seq signal and *IGF1R-AS1*

expression (Pearson correlation > 0.5, *p* < 0.05). Log$_2$FC values of genes and Pearson correlation values of peaks were used to construct edges, replacing RNA-seq Log$_2$FC and ATAC-seq peak heights, respectively.

## Gene activity quantification and survival analysis for the Tempus cohort

**IGF1R-AS1 activity quantification.** Since *IGF1R-AS1* is an unannotated lncRNA not included in the Tempus xR hybrid capture panel, its expression could not be directly measured. To address this, a network activity scoring approach was developed to quantify *IGF1R-AS1* activity based on the expression patterns of genes within its regulatory network. For each sample, expression values of the network genes were normalized by calculating z-scores. A composite score was then generated by summing these z-scores across all network genes. The resulting composite scores were further standardized to yield the final *IGF1R-AS1* network activity score.

**MYC and AR activity quantification.** AR and MYC pathway activity in the Tempus cohort was quantified using a weighted mean approach as previously described, leveraging Hallmark MYC Targets V1 and Hallmark Androgen Response gene sets from the Molecular Signatures Database[85].

**Survival analysis.** Overall survival was calculated from the date of biopsy collection (index date) to either the date of death or last follow-up, with a median follow-up time of 609 days. Patients alive at their last known follow-up were censored at that date. Survival analyses were performed using the R survival package (version 3.7). Initial Cox proportional hazards regression analyses were conducted using penalized splines (pspline) with 4 degrees of freedom to assess the relationship between variables and survival. For visualization and Kaplan–Meier analyses, patients were stratified into high and low groups using optimal cutpoint determination based on maximally selected rank statistics (survminer R package, version 0.4.9). Group differences were assessed using the log-rank test, with statistical significance set at *p* < 0.05. Validation in Taylor et al. 2010 and Abida et al. 2019 cohorts were performed similarly. The 46-gene Prolaris cell-cyle progression signature was obtained from a previous study[27]. Cell-cycle progression score was calculated in the same manner as *IGF1R-AS1* activity score as described above using the 46-gene signature.

## Cell culture

Prostate cancer cell lines VCaP, 22RV1, and LNCaP cells, lung cancer cell line H727, and 293 T were purchased from the American Type Culture Collection (ATCC). VCaP and 293 T cells were cultured in DMEM medium (GenDEPOT, CM002-050) supplemented with 10% fetal bovine serum (FBS, GenDEPOT, F0901-050), and 100 × Antibiotic-Antimycotic (GenDEPOT, CA002-100). 22RV1, LNCaP, and H727 cells were cultured in RPMI1640 medium (GenDEPOT, CM059-050) supplemented with 10% FBS and 100 × Antibiotic-Antimycotic. VCaP cells were precultured in phenol red-free DMEM medium (GenDEPOT, CM004-050) containing 10% charcoal-stripped (CS) FBS (GenDEPOT, F0700-010) for 48 h before DHT treatment. The cells were cultured in a 37 °C incubator and a humidified atmosphere with 5% CO$_2$. All cell lines were authenticated by short tandem repeat (STR) genotyping and were used within 2 months of continuous culturing. Cell lines were Mycoplasma negative as reported by routine laboratory tests.

## Antibodies and reagents

Anti-SMARCA1 (MABE366) and anti-Flag (F1804) were purchased from MilliporeSigma. Anti-GAPDH (sc-47724) was purchased from Santa Cruz Biotechnology. Anti-c-Myc (#18583), anti-Histone 3 (#9715), anti-IGF1R β (#9750), anti-SMARCA1 (#12483), anti-CTCF (#3418), and anti-Flag (#14793) were purchased from Cell Signaling Technology. Anti-

BRD4 (#A301-985A100) and anti-SMC1A (#A300-055A) were purchased from Bethyl Laboratories. Anti-SMARCA4 (ab110641) was purchased from Abcam. All antibodies used for Western blotting analysis were diluted to 1:1000. (+)-JQ1 (S7110) and Enzalutamide (S1250) were purchased from Selleck Chemicals. DHT (A8380) was purchased from Milliporesigma.

## Plasmids, siRNAs, and short hairpin RNA (shRNA) Oligos

Full length and fragments of SMARCA4 and SMARCA1 coding DNA sequences were cloned through PCR amplification from ORF clones of SMARCA4 (RG219258) and SMARCA1 (RC239926) purchased from OriGene and inserted into pCDH-EF1-IRES-eGFP vector. siRNAs targeting *IGF1R-AS1* were purchased from Dharmacon. siRNAs targeting SMARCA1 (SR304469) or SMARCA4 (SR321835) were purchased from OriGene. shRNA oligos targeting *IGF1R-AS1* were constructed in pLKO-vector (#10878) from Addgene. Sequences of siRNA and shRNA oligos are listed in the Supplementary Data 3.

## Lentivirus packaging

Lentivirus was packaged by co-transfection of constructs with second-generation packaging plasmids pMD2.G and psPAX2 into HEK293T cells on 6-well plates. After the first 24 h of transfection (250 ng of pMD2.G, 750 ng of psPAX2, 1 μg of pLKO-shRNA), the medium was changed to DMEM, and 48 and 72 h after transfection, the supernatants were pooled, filtered through a 0.45-μm filter, and used for infection.

## Cell proliferation

Cell proliferation was measured by CellTiter-Glo® 2.0 Cell viability assay (Promega, G9241). Briefly, cells were seeded onto 96-well plates at 3000 cells per well with 100 μl of culture medium. 100 μl of CellTiter-Glo® 2.0 Reagent was added to each well and luminescence signal was recorded after mixing the contents for 15 min on an orbital shaker.

## Migration and invasion assays

Cell migration assays were performed using Transwell chambers (8 μM pore size). Cells in 200 μl serum-free DMEM or RPMI-1640 medium were added to the upper chamber (for LNCaP and 22RV1 cells, $1 \times 10^5$ cells; for VCaP cells, $3 \times 10^5$ cells). In total, 800 μl DMEM medium with 10% FBS was added to the bottom wells of the chambers as a chemo-attractant, and the chambers were incubated at 37 °C with 5% $CO_2$. After 24–48 h, the invaded cells on the bottom surface of the membrane were gently washed with 1× PBS and fixed with 100% methanol for 5 min followed by 0.5% crystal violet staining at room temperature for 15 min. The non-invading cells from the upper surface of the chamber were removed by scrubbing. Randomly, 3–4 fields were captured under microscope and the invaded cell number was counted per field. Similar inserts coated with Matrigel were used to determine cell invasion. Cells in 200 μl serum-free DMEM or RPMI-1640 medium were added to the upper chamber (for LNCaP and 22RV1 cells, $1 \times 10^5$ cells; for VCaP cells, $3 \times 10^5$ cells). A total of 800 μl complete medium was added to the bottom wells of the chambers. Then invaded cell numbers were counted at 24–48 h.

## RNA isolation and quantitative real-time PCR (qRT-PCR)

Total RNA was extracted from cultured cells using the RNeasy Plus Kit (Qiagen, 74134) according to the manufacturer's instructions. cDNA was synthesized from 1 μg of total RNA using the High-Capacity cDNA Reverse Transcription Kit with RNase Inhibitor (ThermoFisher Scientific, 4374966). Samples were amplified for 40 cycles using an Applied Biosystems QuantStudio TM 6 Flex real-time PCR system (Thermo-Fisher Scientific). The $2^{-\Delta\Delta CT}$ method was used to calculate the relative gene expression levels normalized against β-actin. Primer sequences are listed in the Supplementary Data 4.

## RNA immunoprecipitation

Precipitation of RNA bound to target proteins was performed using Magna RIP™ RNA-Binding Protein Immunoprecipitation Kit (Millipore, 17–700). Cell extracts were prepared from subconfluent 15 cm plates, rinsed twice with cold 1× PBS. RIP was performed with whole-cell extracts obtained by shearing $1.0 \times 10^7$ cells in 1 mL of ice-cold RIP Lysis Buffer. All extracts were cleared of debris by centrifugation at 14,000 rpm for 10 min at 4 °C and divided into 100 μL aliquots for different conditions. A 10% volume aliquot was set aside as input. 2–5 μg of isotypic IgG and antibodies against SMARCA1, SMARCA4, CTCF, and Flag-tag were incubated with 50 μL of prewashed protein A + G magnetic beads with rotation for 30 min at room temperature. Then beads-antibody mixture was washed 3 times with 0.5 mL of RIP wash buffer and incubated with 100 μL aliquots of extracts overnight with gentle rotation at 4 °C. After that, beads were washed 6 times with 500 μL of ice-cold RIP wash buffer. Next, 150 μL of proteinase K buffer (1% SDS and 1.2 μg/μl proteinase K in RIP wash buffer) was added to the mixture beads and incubated for 30 min at 55 °C. The supernatant was collected after placing the tubes on the magnetic separator. Then 250 μL of RIP wash buffer and 400 μL phenol:chloroform:isoamyl alcohol (125:24:1) (Sigma-Aldrich, P1944) were added to each tube containing the supernatant. Centrifuge the tubes at 14,000 rpm for 10 min at room temperature and remove 350 μL of the aqueous phase to a new tube. Add 400 μL of chloroform to the aqueous phase. Centrifuge them at 14,000 rpm for 10 min at room temperature to separate the phases. Take 300 μL of the aqueous phase and place it in a new tube. To each tube add 50 μL of ammonium acetate 5 M, 15 μL of LiCl 7.5 M, 5 μL glycogen (5 mg/ml), and 850 μL of absolute ethanol. Keep the samples at −80 °C overnight to allow the RNA precipitation, then centrifuge at 14,000 rpm for 30 min at 4 °C and discard the supernatant. Wash the pellet with 500 μL of cold 80% ethanol and centrifuge at 14,000 rpm for 15 min at 4 °C. Discard the supernatant and air-dry the pellets. Resuspend the pellets in 20 μL of RNase-free water. RNA was reversely transcribed and the enrichment of *IGF1R-AS1* and *SChLAP1* was measured with specific primers by qPCR (Supplementary Data 4).

## In vitro RNA transcription

The *IGF1R-AS1* cDNA was cloned into pCDH-EF1-IRES-eGFP vector. Biotin-labeled RNAs were transcribed in vitro by T7 RNA polymerase (Promega, P2075) with biotin RNA labeling mix (Roche, 11685597910), using T7-DNA PCR product as templates. After treatment with DNase I, transcribed RNAs were purified by RNeasy MinElute Cleanup Kit (Qiagen, 74204). Then RNA size and purity were verified through running RNA gel.

## LncRNA secondary structure prediction

The secondary structure of lncRNA *IGF1R-AS1* was predicted via the RNAfold web server[86] (available at http://rna.tbi.univie.ac.at/cgi-bin/RNAWebSuite/RNAfold.cgi). The *IGF1R-AS1* RNA sequence was processed using the default parameters to determine the structure with the minimum free energy (MFE).

## RNA pulldown and protein mass spectrometry

Proteins bound to *IGF1R-AS1* were identified by mass spectrometry. In summary, nuclear extracts were prepared from around $1 \times 10^7$ VCaP cells using NE-PER™ Nuclear and Cytoplasmic Extraction Reagents (Thermo Scientific, 78833). Protein concentration of nuclear extract was measured by Bradford assay, and the nuclear extract was diluted with RIP wash buffer to ~2 mg/ml. The nuclear extract was precleared with streptavidin T1 magnetic beads (Invitrogen, 65601) in RIP wash buffer for 1 h with rotation at room temperature. After that, a 10% volume aliquot was set aside as input. Then, 50 pmol of in vitro transcribed biotinylated RNA was incubated with 650 μL of 0.1 μg/μL yeast tRNA-supplemented RIP wash buffer and 650 μL of the precleared

nuclear extract overnight at 4 °C with rotation. Next, 60 µL of pre-washed streptavidin beads were added to the mix and incubated for one extra hour. Finally, the beads were then washed 5 times with 500 µL of RIP wash buffer. In the last wash, all the supernatant was carefully removed, and the beads were resuspended in protein sample buffer, boiled at 95 °C for 10 min, loaded and ran in precast 4–12% Bis-Tris gel for several minutes. Protein gels were cut and sent to Northwestern University Proteomics Facility for protein mass spectrometry analysis.

### In vitro RNA-protein binding assay
5 pmol of biotinylated RNA was incubated with different amounts of recombinant human SMARCA1 protein (OriGene, TP318308) or SMARCA4 protein (Abcam, ab82237) for 1 hr at RT in 200 µl of binding buffer (50 mM Tris-HCl pH 7.9, 10% glycerol, 100 mM KCl, 5 mM MgCl2, 10 mM β-ME, 0.1% NP-40, 1 mM PMSF, 1 × Superase-in, and 1 × protease inhibitor cocktail). Then, 30 µl of washed streptavidin-conjugated magnetic beads were added to each reaction, and the mixtures were incubated at RT for 30 min. Beads were washed five times and boiled in 1 × SDS loading buffer, and the retrieved protein was analyzed using western blotting.

### Protein isolation and Western blotting
Cells were lysed in mammalian protein extraction reagent (Pierce, 89900). After quantification using a BCA protein assay kit (Pierce, 23225), total proteins were separated by SDS-PAGE under denaturing conditions and transferred to PVDF membranes (Bio-Rad, 162-0177). Membranes were blocked in 5% non-fat milk (Bio-Rad, 1706404) and then incubated with primary antibodies overnight at 4 °C, followed by incubation with goat anti-mouse/rabbit/rat IgG (H + L)-HRP secondary antibody (GenDEPOT, SA002-500, SA001-500, SA006-500, diluted to 1:4000). Finally, the protein bands were immunodetected by ECL blotting substrates (Bio-Rad, 1705060) and captured by the ChemiDoc Imaging System (Bio-Rad, Hercules, CA).

### Single-molecule RNA FISH and confocal microscopy
Single-molecule RNA fluorescence in situ hybridization (smFISH) was performed to detect *IGF1R-AS1* transcripts using custom-designed Stellaris FISH probes (LGC Biosearch Technologies). VCaP cells ($2 \times 10^6$) were seeded on 12 mm coverslips 24 h before hybridization. Cells were fixed with 4% paraformaldehyde (10 min), quenched with 0.1 M glycine (10 min), and washed with PBS. After permeabilization with 0.5% Triton ×-100 (30 min), cells were incubated in pre-hybridization buffer (10% formamide/2× SSC) for 30 min. RNasin (Promega, N2111) was included throughout to prevent RNA degradation.

Hybridization was performed for 3 h at 37 °C in buffer containing 2× SSC, 10% formamide, 10% dextran sulfate, 0.2 mg/mL salmon sperm ssDNA, 2 mg/mL BSA, 100 nM FISH probes, and 40U RNasin. After two 20-min washes (37 °C) with 10% formamide/2× SSC and a final PBS wash, coverslips were mounted using prolonged diamond anti-fade medium. Probe specificity was validated using siRNA-mediated knockdown of *IGF1R-AS1*.

Imaging was performed on a custom Nikon microscope equipped with a Yokogawa W1 spinning disk, Kinetix CMOS camera (Photometrics), 60× oil immersion objective (NA 1.45), and lasers (405, 488, 561, 640 nm). DAPI and ATTO647N-labeled *IGF1R-AS1* probes were sequentially excited using 405 nm and 640 nm lasers. Analysis was conducted using FIJI ImageJ.

### Chromatin immunoprecipitation (ChIP)
Precipitation of DNA bound to target proteins was performed using Magna ChIP™ A/G Chromatin Immunoprecipitation Kit (Millipore, 17-10085). Cells were cross-linked in 1% formaldehyde at room temperature for 10 min. Crosslinking was terminated by the addition of 125 mM

(final concentration) glycine. Cells were then incubated with glycine at room temperature for 5 min with gentle mixing, washed with ice-cold PBS, scraped, and resuspended in PBS. After washing, cell pellets were lysed using cell lysis buffer (10 mM Tris-HCl [pH 7.5], 10 mM NaCl, 3 mM MgCl2, 0.5% IGEPAL, 1 mM PMSF), incubated on ice for 10 min, and then centrifuged at 800 g for 5 min at 4 °C to pellet the nuclei. Pellets were lysed using nuclear lysis buffer (50 mM Tris-HCl [pH 8.0], 10 mM EDTA, 1% SDS, 1 mM PMSF, and proteinase inhibitor cocktail) and then diluted with IP dilution buffer (16.7 mM Tris-HCl [pH 8.0], 1.2 mM EDTA, 1.1% Triton ×-100, 167 mM NaCl, 0.01% SDS). Nuclear lysates were sonicated, and the debris was removed by centrifugation. 2–5 µg of Anti-BRD4, anti-AR, anti-SMARCA1, anti-SMARCA4, anti-CTCF, anti-SMC1A, and IgG antibodies were mixed with clear nuclear lysates for immunoprecipitation. Co-precipitated DNA was purified, and the enrichment of target DNA fragments was measured with specific primers by qPCR. ChIP-qPCR primers used in this study are listed in Supplementary Data 4.

### Quantitative analysis of chromosome conformation capture assays (3C-qPCR)
3C-qPCR was performed following a published protocol with minor modifications[87]. Briefly, cells under various treatments were crosslinked with 1% formaldehyde for 10 min at room temperature, and the reaction was quenched with 0.125 M glycine. Nuclei were isolated, lysed, and digested overnight at 37 °C with 100 U of *MluCI* (NEB, R0538) under rotation. Ligation was carried out at 16 °C for 4 h, followed by a 30-min incubation with 5 µl of T4 DNA ligase (NEB, M0202) at room temperature. Reverse crosslinking was performed overnight at 68 °C in the presence of proteinase K. Genomic DNA was extracted using phenol-chloroform. After treatment with 1 µl of RNase A (Invitrogen, AM2270) for 30 min at 37 °C, secondary digestion was performed with 100 U of *HindIII* (NEB, R0104S) for 2 h at 37 °C under rotation, followed by a second phenol-chloroform extraction. For qPCR analysis, GAPDH served as an internal reference, and relative interaction frequencies were calculated using the ΔΔCt method, normalized to the E3 region. Primer sequences are provided in Supplementary Data 4.

### Xenograft mouse model
CB-17 SCID male mice (strain 236, ages 6–7 weeks) and NOD-SCID NCG male mice (strain 572, ages 5–6 weeks) were purchased from Charles River. All animal care and experimental procedures were conducted in compliance with institutional and National Institutes of Health (NIH) guidelines, with approval from the University of Minnesota and Northwestern University Institutional Animal Care and Use Committee (IACUC). All mice were housed in a specific pathogen-free facility maintained at 21.5 ± 1 °C and 30–70% humidity, with a 12-h dark-light cycle and access to food and water. Animal health and behavior were monitored daily throughout the study. Mice were weighed twice weekly to monitor for cachexia or rapid weight loss. For VCaP xenograft experiments, $2 \times 10^6$ VCaP cells expressing *IGF1R-AS1* shRNA or control shRNA were suspended in 100 µl PBS containing 50% Matrigel and implanted subcutaneously into the dorsal flank of pre-castrated CB-17 SCID male mice. For 22RV1 xenograft experiments, $2 \times 10^6$ 22RV1 cells expressing *IGF1R-AS1* shRNAs or control shRNA were suspended in 100 µl PBS containing 50% Matrigel and implanted subcutaneously into the dorsal flank of pre-castrated NOD-SCID NCG male mice. Tumor volumes were measured by calipers for length (a), width (b), and calculated as tumor volume = MIN(a)^2 × MAX(b)/2. Once the average tumor volume in the control group reached 2000 mm³, mice were euthanized via CO₂ inhalation. Mice's death was subsequently confirmed by cervical dislocation as a secondary physical method in accordance with institutional guidelines. Additional criteria for early euthanasia included a body weight loss exceeding 10%, tumor ulceration, or any tumor growth that impeded normal physiological function.

## Statistical analysis

Statistical analysis was carried out using the GraphPad Prism 10 to assess differences between experimental groups. Statistical differences were analyzed by two-tailed Student's t-test, Chi-square test, or one-way and two-way ANOVA where appropriate. p values lower than 0.05 were considered to be statistically significant. No statistical methods were used to predetermine sample sizes, but our sample sizes are similar to those reported in previous studies. The results were reproducible and conducted with established internal controls. When feasible, experiments were repeated two or more times and yielded similar results.

## Reporting summary

Further information on research design is available in the Nature Portfolio Reporting Summary linked to this article.

## Data availability

*Data generated in this study:* Raw and processed data generated in this study have been deposited in the GEO repository (https://www.ncbi.nlm.nih.gov/geo/) under accession number **GSE208745**. This includes RNA-seq data for *IGF1R-AS1*, *SMARCA1*, and *SMARCA4* knockdown experiments with controls in VCaP cells, ATAC-seq data for *IGF1R-AS1* knockdown with controls, SMARCA1 knockdown with controls in VCaP cells, CTCF HiChIP data for *IGF1R-AS1* knockdown with controls in VCaP cells, and PacBio Iso-seq data for *IGF1R-AS1* 3′ RACE products in VCaP and H727 cells. The mass spectrometry proteomics data are available via ProteomeXchange (http://www.proteomexchange.org/) under accession number **PXD074272**. *Third-party/restricted data:* The RNA-seq and associated de-identified clinical annotation data from 499 CRPC patients used in this study were obtained from Tempus AI, Inc. under a research collaboration agreement. These data constitute restricted-access third-party clinical data and were not generated as part of the current study. Due to patient privacy considerations, institutional review board requirements, and commercial data-use agreements, individual-level raw data cannot be made publicly available. Access to the Tempus data may be granted to qualified academic or clinical researchers subject to approval by Tempus AI, Inc. and execution of an appropriate data-use agreement. The authors do not have the authority to grant access to these data. *Publicly available repositories*: • **SU2C cohort**: PolyA-enriched RNA-seq data from 101 metastatic prostate cancer samples, available at dbGaP (https://dbgap.ncbi.nlm.nih.gov/home/) (accession: **phs000915.v2.p2**). • **TCGA cohort**: Normalized ATAC-seq read counts and normalized RNA-seq gene expression values (FPKM-UQ) were obtained from the Xena Browser (https://xenabrowser.net/datapages/). • **Cell line RNA-seq**: ○ VCaP, MDA-PCa-2b, 22RV1, LNCaP, DU145, NCI-H660, and PC3 (NCBI SRA under accessions: **SRX5417211**, **SRX5414821**, **SRX5414881**, **SRX5414853**, **SRX5414453**, **SRX5414893**, **SRX5414759**). ○ H727 cells treated with JQ1 and DMSO were obtained through prior communication with study authors. ○ RNA-seq for VCaP with DMSO (24 h) and AU-15330 (4 h) (NCBI SRA under accessions: **SRX10515156**, **SRX10515157**, **SRX10515158**, **SRX10515159**). ○ Nanopore direct RNA-seq for VCaP (NCBI SRA under accession: **SRX26188102**). • **Histone ChIP-seq**: ○ H3K4me3 data for LNCaP, K562, HepG2, HeLa-S3, and GM12878 (ENCODE project (https://www.encodeproject.org/) under accessions: **ENCSR000DWF, ENCSR668LDD, ENCSR575RRX, ENCSR000AOF, ENCSR057BWO**). ○ H3K27ac data for VCaP (ENCODE project under accession: **ENCSR597ULV**). ○ PolII data for VCaP (NCBI SRA under accession: **SRX471863**). ○ H3K4me3 data for VCaP (NCBI SRA under accession: **SRX022554**). • **TF ChIP-seq, HiChIP and ATAC-seq**: ○ BRD4 and PolII data for VCaP (GEO under accession: **GSE148358**). ○ SMARCA4 data for VCaP with EtOH treatment, siS-MARCA4 ATAC-seq with EtOH treatment, and corresponding controls (GEO under accession: **GSE136016**). ○ SMARCA4 data for 22RV1 and

LNCaP (NCBI SRA under accessions: **SRX4193367**, **SRX2545045**). ○ ATAC-seq for VCaP with DMSO treatment (24 h) and AU-15330 treatment (4 h) (NCBI SRA under accessions: **SRX10525411**, **SRX10525413**; peaks: GEO under accession: **GSE171584**). ○ H3K27ac and CTCF HiChIP-seq and ChIP-seq for VCaP with AU-15330 (4 h) and DMSO (24 h) (GEO under accession: **GSE171591**). • **ChIA-PET**: PolII data for VCaP, LNCaP, and DU145 (GEO under accession: **GSE121020**). • **CRPC model data**: RNA-seq and ATAC-seq (GEO under accession: **GSE199190**). • **BigWig files**: CTCF and RAD21 ChIP-seq data (ENCODE project under accessions: **ENCFF507CRU**, **ENCFF704QYE**, **ENCFF539QXW**, **ENCFF336UPT**, **ENCFF083AEY**, **ENCFF341HKN**, **ENCFF543QNU**, **ENCFF867GSQ**, **ENCFF653EFX**, **ENCFF775EKJ**). The remaining data are available within the Article, Supplementary Information or Source Data file. Source data are provided with this paper.

## Code availability

The source codes for lncRNA identification and gene-peak regulatory network construction are available at https://github.com/ylab-hi/trans_lncRNA.

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

## Acknowledgements

We gratefully acknowledge Drs. Kate Lines and Rajesh Thakker from the University of Oxford for sharing the RNA-seq data of H727 cells treated with JQ1. We also thank the University of Minnesota Genomics Center for providing PacBio Iso-seq sequencing services. Proteomics services were performed by the Northwestern Proteomics Core Facility (RRID:SCR_017945), generously supported by NCI CCSG P30 CA060553 awarded to the Robert H Lurie Comprehensive Cancer Center and the National Center for Translational and Developmental Proteomics supported by RM1 GM156535. This work was supported in part by grants from the National Institutes of Health (R01CA259388 and R35GM142441 to R.Y.; R01CA256741, R01CA278832 and R01CA285684 to Q.C.; R01CA300246 and Prostate SPORE P50CA180995 Developmental Research Program to Q.C. and R.Y.; R01CA174777, R01CA270539 and R01CA276269 to S.M.D.; R35GM150941 to Y.L.), the Department of Defense (W81XWH-21-1-0146 and HT9425-23-1-0491 to Y.Y.; W81XWH-19-1-0563 and W81XWH-20-1-0504 to Q.C.), the Prostate Cancer Foundation Young Investigator Award to R.Y., and the Polsky Urologic Cancer Institute of the Robert H. Lurie Comprehensive Cancer Center of Northwestern University at Northwestern Memorial Hospital to Q.C. and R.Y., the American Cancer Society (Research Scholar Grant RSG-21-034-01-TBG to L.H.H.), the Lung Cancer Research Foundation (992504 to L.H.H.), and the Windfeldt Cancer Research Award to L.H.H.

## Author contributions

R.Y., Q.C., and S.M.D. conceptualized and designed the study. R.Y. identified the *IGF1R-AS1* lncRNA. Y.Y. and Y.L. (Yingming Li) led the experiments and data acquisition, with contributions from Q.M., Q.G., N.E.P., A.R., K.H.W., Q.L., C.D., P.I., X.Z. (Xiaoyang Zhang), S.K.A., L.H.H., X.Z. (Xihong Zhang), D.Y., and Y.L. (Yang Liu). Y.L. (Yingming Li), Y.Y., X.Z. (Xingxing Zhang), and T.A. conducted the mouse xenograft studies. T.-Y.W. and J.F. led the computational analyses with input from Y.R., S.F., A.B.W., and E.M.S. Y.Y., T.-Y.W., J.F., S.M.D., Q.C., and R.Y. collaboratively wrote the manuscript. All authors reviewed the results and provided feedback on the manuscript.

## Competing interests

Rendong Yang has served as an advisor/consultant for Tempus AI, Inc. This relationship did not influence the research presented in this study. The remaining authors declare no competing interests.

## Additional information

[1]Department of Urology, Northwestern University Feinberg School of Medicine, Chicago, IL, USA. [2]Bioinformatics and Computational Biology Program, University of Minnesota, Minneapolis, MN, USA. [3]Masonic Cancer Center, University of Minnesota, Minneapolis, MN, USA. [4]Department of Biochemistry, University of Utah, Salt Lake City, UT, USA. [5]Department of Oncological Sciences, Huntsman Cancer Institute, University of Utah, Salt Lake City, UT, USA. [6]Department of Pharmacology, Northwestern University Feinberg School of Medicine, Chicago, IL, USA. [7]Tempus AI, Inc., Chicago, IL, USA. [8]The Hormel Institute, University of Minnesota, Austin, MN, USA. [9]Department of Urology, Cedars Sinai Medical Center, Los Angeles, CA, USA. [10]Departments of Laboratory Medicine and Pathology and Urology, University of Minnesota, Minneapolis, MN, USA. [11]Robert H. Lurie Comprehensive Cancer Center, Northwestern University Feinberg School of Medicine, Chicago, IL, USA. [12]Present address: Cancer Research Program, Research Institute of the McGill University Health Centre, Montreal, QC, Canada. [13]These authors contributed equally: Yongyong Yang, Ting-You Wang, Joshua Fry, Yingming Li, Qingshu Meng. ✉e-mail: dehm@umn.edu; qi.cao@northwestern.edu; rendong.yang@northwestern.edu

