## [Transparent Peer Review file · Nature Communications]

Tumor-specific lncRNA IGF1R-AS1 trans-regulates chromatin interactions associated with oncogenic MYC signaling

Corresponding Author: Dr Rendong Yang

Version 0:

Reviewer comments:

Reviewer #1

(Remarks to the Author)

Yang et al identified 1,344 new lncRNAs in prostate cancer data sets, and performed further bioinformatics analyses and experiments to show lncRNA IGF1R-AS1 interacts with SWI/SNF complex to regulate MYC expression and cell proliferation in VCaP cells. The data are well-presented, and the manuscript is clearly written and well-organized. However, I have several suggestions to improve the novelty, mechanistic insight, and robustness of this study.

1. The authors have performed 3' RACE and PacBio Iso-seq to validate IGF1R-AS1 isoforms. Since this is a novel lncRNA, additional Northern blotting and 5' RACE would further confirm transcript integrity and size.
2. The functional characterization of IGF1R-AS1 is currently limited to the VCaP cell line. Validation in more PCa cell lines is necessary for both in vitro and in vivo assays. Additionally, for the in vivo tumor growth assay, they only used one IGF1R-AS1 shRNA (Fig 3h, shAS1-2#). shAS1-1# should also be used for in vivo tumor growth assay.
3. This study reports an interaction between IGF1R-AS1 and the SWI/SNF complex. However, given the extensive literatures on lncRNA-protein interactions in cancer, including with the SWI/SNF complex, the conceptual advances here appear incremental. The authors should provide a detailed analysis of this interaction to identify the specific protein domains and RNA sequences involved.
4. The mechanism, by which the SWI/SNF-IGF1R-AS1 complex targets specific enhancers or promoters genome-wide, remains unclear. Since the key novelty of this study is the cooperation of SWI/SNF and IGF1R-AS1 to regulate epigenome, clarification of the mechanism is necessary.
5. They claimed IGF1R-AS1 modulates CTCF-mediated chromatin loops, but the current data are insufficient to support a robust conclusion. There is no CTCF HiChIP data to show reduced CTCF looping upon IGF1R-AS1 knockdown, and the mechanism explaining why knockdown decreases CTCF binding or looping is not addressed. The mechanism here is also related with the genome-wide specific targeting of SWI/SNF-IGF1R-AS1 complex.
6. The development of new targeting drugs based on this lncRNA, with thorough in vivo assessments of identity, PK, toxicity and efficacy, should improve the translational impact of this study.
7. Since IGF1R-AS1 was identified in metastatic castration-resistant prostate cancer (mCRPC) datasets, its potential role in resistance to AR-targeting therapies warrants investigation. In vivo assays addressing this question could increase the clinical importance of this study.

(Remarks on code availability)

N/A

Reviewer #2

(Remarks to the Author)

The authors discover the antisense lncRNA IGF1R-AS1, transcribed from a high-density super-enhancer within the IGF1R locus, and delineate its oncogenic mechanism in metastatic castration-resistant prostate cancer (mCRPC). IGF1R-AS1 recruits SWI/SNF chromatin-remodelling and CTCF/cohesin architectural complexes to establish long-range loops that juxtapose distal MYC enhancers with the MYC promoter, thereby amplifying MYC expression and tumorigenicity. The work illustrates how a super-enhancer-driven lncRNA can remodel three-dimensional genome topology in trans to activate distal oncogenic circuitry in a cancer-specific context.

1. Threshold for lncRNA inclusion (Fig. 1a).

The manuscript states that lncRNAs retained for downstream analysis are those expressed in ≥ 5 of 101 mCRPC specimens. Please justify this 5-sample threshold. Does it represent a statistical cut-off (e.g. to control the false-discovery rate) or a biological rationale (e.g. prevalence $\geq 5\%$)?

2. Please describe explicitly how many of the 101 tumors express IGF1R-AS1 above background and provide the expression distribution (e.g. violin plot or histogram).

3. Most functional assays are performed in VCaP cells. To strengthen the claim that IGF1R-AS1 is broadly oncogenic in mCRPC, please test at least one additional metastatic line with high endogenous expression (e.g., MDA-PCa-2b or 22Rv1) using knockdown. Conversely, introduce IGF1R-AS1 into a line (e.g., LNCaP) to determine whether overexpression is sufficient to confer the oncogenic phenotypes.

4. Selection of TCGA specimens (Fig. 2b). Only three "positive" TCGA prostate samples are shown. Clarify the selection criteria ("positive" is ambiguous). Were these the top three expressers of IGF1R-AS1, or the only cases with matched normal tissue?

5. The author described that both SMARCA1 and SMARCA4 interact with the MYC promoter and enhancer regions and bind IGF1R-AS1. However, the subsequent analyses rely exclusively on SMARCA4 ChIP-seq data. Are the authors focusing solely on SMARCA4 because SMARCA1 and SMARCA4 operate within the same SWI/SNF chromatin-remodeling complex? Please clarify this point in the manuscript.

Minor comments

Typographical error (Fig. 2h). "siNON" should read "siCon" (scramble control).

Labelling ambiguity (Fig. 3h). The axis label "siAS-1 #1/#2 meta" is unclear. If both siRNAs were co-transfected, please indicate "mix" or "pool".

(Remarks on code availability)

Reviewer #3

(Remarks to the Author)

(Remarks on code availability)

Reviewer #4

(Remarks to the Author)

In this manuscript, Yang et al. identify IGF1R-AS1, a previously uncharacterized tumor-specific lncRNA, as a key regulator in metastatic castration-resistant prostate cancer (mCRPC). Through integrative multi-omics analyses, the authors demonstrate that IGF1R-AS1 interacts with the chromatin remodelers SMARCA1 and SMARCA4 to orchestrate long-range chromatin looping between the distal MYC enhancer E3 and its promoter, thereby amplifying MYC expression and driving tumorigenesis. This work not only unveils a novel trans-acting mechanism by which lncRNAs modulate oncogenic signaling but also highlights the broader potential of tumor-specific lncRNAs as therapeutic targets in cancer. This is a well-executed study with significant findings. However, some mechanistic details require further refinement before publication. Below is a detailed review with suggestions for further improvement:

1. The manuscript suggests IGF1R-AS1 recruits SWI/SNF to MYC enhancers, but the exact binding sites or motifs mediating this interaction are unclear. Additional experiments (e.g., truncation mutants of IGF1R-AS1 to identify functional domains) could strengthen this claim. Moreover, it is interesting to see whether IGF1R-AS1 directly binds SMARCA1 and SMARCA4. Can these two proteins directly bind to IGF1R-AS1 *in vitro*? If so, which functional domains are responsible for the direct interaction?

2. As CCAT1 is localized in a super-enhancer region, it is extensively transcribed in many types of cancers and has been demonstrated to have a 5' additional extension in HeLa cells (see Cai et al., Nature, 2020), I am wondering whether IGF1R-AS1 RNA directly interact with the enhancer RNAs transcribed in E3 region? could they form RNA base pairings? This can explain why IGF1R-AS1 specifically targets the chromatin remodeling complex to this enhancer and potentially to thousands of other sites.

3. While the study shows IGF1R-AS1 does not regulate IGF1R, it would be helpful to explicitly test whether IGF1R-AS1 knockdown affects other nearby genes (e.g., by RNA-seq or qPCR).

4. For the IGF1R-AS1 knockdown studies, it is better to reintroduce IGF1R-AS1 in knockdown cells (e.g., via overexpression) to confirm that the observed phenotypes are due to IGF1R-AS1 loss and not off-target effects.

5. The Tempus cohort analysis suggests IGF1R-AS1 activity predicts poor survival. Expanding this to other cohorts or validating in independent datasets would strengthen the biomarker claim.

6. Several typos need to be fixed: Page 3: "evolutionally" → "evolutionarily"; Page 9: "fueling cancer progression" → "driving

cancer progression"

(Remarks on code availability)

Version 1:

Reviewer comments:

Reviewer #1

(Remarks to the Author)

The authors have comprehensively addressed my concerns. The incorporation of Nanopore direct RNA sequencing, the validation in a second CRPC cell line (22RV1), the detailed domain mapping of the lncRNA-protein interactions, and the CTCF HiChIP analysis strengthened the novelty, mechanistic depth, and robustness of the study. I have no further comments.

(Remarks on code availability)

Reviewer #2

(Remarks to the Author)

Overall, the authors' responses are clear, well-reasoned, and comprehensive, and the issues raised have been well addressed. I have minor comments for the paper.

1. Introduction: 'Opening new avenues for therapeutic intervention' feels too ambitious given the current data. The study compellingly demonstrates tumor-specific lncRNA-mediated distal regulation and 3D chromatin architecture, but the evidence does not yet support clinical or therapeutic claims.
2. Response 7. The current distribution plot is not straightforward for evaluating prevalence. The authors' point is that the IGF1R-AS1 junction is more prevalent in metastatic vs localized tumors, showing raw sample counts obscure the comparison because the metastatic (Met) and localized (PCa) cohorts differ in size. A pie chart summarizing the proportions should be sufficient
3. Following the authors' clarification of the rationale for studying SMARCA4, I recommend inserting concise biological relevance and functional alignment justifications to strengthen clarity and coherence.

(Remarks on code availability)

Reviewer #3

(Remarks to the Author)

(Remarks on code availability)

Reviewer #4

(Remarks to the Author)

The revised manuscript has fully addressed all of my concerns, and I am pleased to support its publication.

(Remarks on code availability)

Point-by-Point response

RE: **NCOMMS-25-35735** - Tumor-specific lncRNA IGF1R-AS1 trans-regulates chromatin interactions associated with oncogenic MYC signaling

We would like to thank the editor for the opportunity to allow us to revise our manuscript and thank all reviewers for your insightful and constructive comments, which have significantly improved the quality of our work. We have carefully considered the comments and are committed to addressing each one of them. As requested, we have added new data and rewritten some parts of the manuscript. We believe that these changes have resulted in a more solid and cohesive manuscript. We will now provide a point-by-point response to the reviewers' comments. The changes in the manuscript are highlighted in red. We hope the Reviewers and the editor will find this manuscript to be much improved and suitable for publication. We want to assure the reviewers and the editor that we take their comments seriously and are dedicated to addressing them to the best of our ability.

REVIEWER COMMENTS

Reviewer #1, expertise in prostate cancer epigenetics, lncRNAs and MYC (Remarks to the Author):

Yang et al identified 1,344 new lncRNAs in prostate cancer data sets, and performed further bioinformatics analyses and experiments to show lncRNA IGF1R-AS1 interacts with SWI/SNF complex to regulate MYC expression and cell proliferation in VCaP cells. The data are well-presented, and the manuscript is clearly written and well-organized. However, I have several suggestions to improve the novelty, mechanistic insight, and robustness of this study.

Response: We greatly appreciate the reviewer's supportive evaluation of our manuscript.

1. The authors have performed 3' RACE and PacBio Iso-seq to validate IGF1R-AS1 isoforms. Since this is a novel lncRNA, additional Northern blotting and 5' RACE would further confirm transcript integrity and size.

Response: We sincerely thank the reviewer for this crucial suggestion and agree that further confirmation of IGF1R-AS1 transcript integrity and size is essential.

Northern blot is a traditional method for determining the size and integrity of RNA transcripts; however, it has several significant disadvantages, such as low sensitivity and the potential for hazardous radioactive probes. Therefore, we adapted to use the third-generation long-read sequencing technology, Nanopore direct RNA sequencing, which preserves the native state of RNA and sequences entire RNA molecules from end to end, providing more accurate RNA size

measurements, and its real-time sequencing allows direct assessment of RNA integrity from the sample, and PacBio long-reads sequencing, which can also accurately sequence the full-length of transcripts.

To directly address the concern, we analyzed our whole-transcriptome Nanopore direct RNA sequencing (dRNA-seq) data generated from VCaP cells, which was used in a separate study¹ and deposited into GEO under accession GSE277934. We found that the major IGF1R-AS1 isoform size and sequence in VCaP cells is consistent with the analysis from PacBio Iso-seq data in VCaP cells. This compelling evidence confirms the integrity and size of the IGF1R-AS1 transcript, and the result has been incorporated into **new supplementary Fig. 2b (line 180-181)** (and shown in **Response Fig. 1** for reference).

2. The functional characterization of IGF1R-AS1 is currently limited to the VCaP cell line. Validation in more PCa cell lines is necessary for both *in vitro* and *in vivo* assays. Additionally, for the *in vivo* tumor growth assay, they only used one IGF1R-AS1 shRNA (Fig 3h, shAS1-2#). shAS1-1# should also be used for *in vivo* tumor growth assay.

Response: We thank the reviewer for these crucial suggestions regarding the broader validation. We conducted extensive functional characterization of IGF1R-AS1 in a second PCa cell line, 22RV1, which expresses detectable IGF1R-AS1. We performed IGF1R-AS1 knockdown using two independent siRNAs or shRNAs in 22RV1 cells and then evaluated cell growth, migration, and invasion. Consistent with the findings in VCaP cells, knocking down IGF1R-AS1 in 22RV1 cells suppressed cell growth, migration, and invasion (**new supplementary Fig. 3b-d**). In addition, we performed *in vivo* murine xenograft assays using 22RV1 cells with IGF1R-AS1 stable knockdown by two different shRNAs (shAS1-1# and shAS1-2#), and observed that both IGF1R-AS1 shRNAs significantly inhibited xenograft tumor progression *in vivo* (**new supplementary Fig. 3k-n**).

These new data have been incorporated into **revised supplementary Fig. 3 (lines 243-246, and lines 252-254)** (and shown here in **Response Fig. 2** for reference).

3. This study reports an interaction between IGF1R-AS1 and the SWI/SNF complex. However, given the extensive literatures on lncRNA-protein interactions in cancer, including with the SWI/SNF complex, the conceptual advances here appear incremental. The authors should provide a detailed analysis of this interaction to identify the specific protein domains and RNA sequences involved.

Response: We thank the reviewer for this critical suggestion. We have performed the RNA-protein domain mapping to identify the specific regions governing the interaction between IGF1R-AS1 with SMARCA1 and SMARCA4.

First, to investigate whether IGF1R-AS1 directly interacts with SMARCA1 and SMARCA4, we performed *in vitro* RNA-protein binding assay with *in vitro*-transcribed biotin-labeled IGF1R-AS1 RNA and purified recombinant His-tagged SMARCA4 protein, or Flag-tagged SMARCA1 protein, which demonstrated that IGF1R-AS1 binds directly to SMARCA1 protein and SMARCA4 protein (new Fig. 4f-g).

To map the IGF1R-AS1 functional motifs corresponding to SMARCA1 and SMARCA4 binding, we predicted the secondary structure of IGF1R-AS1 using RNAfold webserver (new Fig. 4h), and conducted *in vitro* RNA pull-down assays using a series of truncated biotin-labeled IGF1R-AS1 fragments (new Fig. 4i). This analysis revealed that nucleotides 512-645 of IGF1R-AS1 are critical for the interaction with SMARCA4 protein, while nucleotides 177-407 and 408-511 of IGF1R-AS1 are essential for the interaction with SMARCA1 protein.

Both SMARCA1 and SMARCA4 contain an ATPase domain in the middle of the protein sequence (new Fig. 4j-k). We cloned the full-length, N-terminal domain, ATPase domain, and C-terminal domain of both SMARCA1 and SMARCA4, each with a C-terminal Flag tag. We expressed them in VCaP cells and then performed RIP assay using anti-Flag antibody (new supplementary Fig. 4g-h), followed by qPCR to analyze the enrichment of IGF1R-AS1 by different protein domains (new Fig. 4j-k). The results showed that IGF1R-AS1 mainly binds to the ATPase domain of SMARCA4 and the C-terminal domain of SMARCA1 (new Fig. 4j-k).

These detailed mapping studies significantly strengthen our findings by identifying the specific RNA motifs and protein domains responsible for the IGF1R-AS1-SMARCA4 and IGF1R-AS1-SMARCA1 interaction. These new data have been incorporated into revised Fig. 4 and supplementary Fig. 4 (lines 292-302) (see Response Fig. 3 for reference).

4. The mechanism, by which the SWI/SNF-IGF1R-AS1 complex targets specific enhancers or promoters genome-wide, remains unclear. Since the key novelty of this study is the cooperation of SWI/SNF and IGF1R-AS1 to regulate epigenome, clarification of the mechanism is necessary.

Response: We thank the reviewer for this crucial comment.

Previous investigations demonstrated that CTCF and SMARCA4 physically interact and colocalize to enhancers and regulate gene expression²⁻⁴. In our analysis, we showed notably disrupted CTCF binding and reduced CTCF-mediated chromatin interactions around the E3 region in VCaP cells after the treatment with SWI/SNF degrader (Fig. 6b). Our IGF1R-AS1 pulldown protein MS data also suggest that CTCF binds IGF1R-AS1, and thus it is very likely that IGF1R-AS1 regulates the epigenome by affecting the interaction between CTCF and SMARCA4. We validated the interaction between IGF1R-AS1 and CTCF through CTCF RIP-qPCR (new Fig. 6g) and IGF1R-AS1 pulldown assay, which further showed that IGF1R-AS1 interacts with CTCF through nucleotides 177-407 (new Fig. 6h), while it interacts with SMARCA4 through nucleotides 512-645 (new Fig. 4i, See Response Fig. 3). We further performed CTCF co-IP in VCaP cells and found that knocking down IGF1R-AS1 reduced the interaction between SMARCA4 and CTCF (new Fig. 6i), indicating that IGF1R-AS1 acts as a scaffold to enhance their interaction and further regulate the epigenome. We have added these results into revised Fig. 4 and Fig. 6 (lines 446-453) (see Response Fig. 4 for reference).

5. They claimed IGF1R-AS1 modulates CTCF-mediated chromatin loops, but the current data are insufficient to support a robust conclusion. There is no CTCF HiChIP data to show reduced CTCF looping upon IGF1R-AS1 knockdown, and the mechanism explaining why knockdown decreases CTCF binding or looping is not addressed. The mechanism here is also related with the genome-wide specific targeting of SWI/SNF-IGF1R-AS1 complex.

Response: We thank the reviewer for this important comment.

We performed CTCF HiChIP in siRNA negative-control and IGF1R-AS1 knockdown VCaP cells to directly assess the impact of IGF1R-AS1 on CTCF-mediated chromatin looping. This new data was incorporated into the revised Fig. 6. Analysis of this CTCF HiChIP data confirmed down-regulation of the E3-MYC promoter loop upon IGF1R-AS1 depletion (new Fig. 6d, see Response Fig. 5). Importantly, aggregate peak analysis (APA) showed significant reduction in CTCF looping strength genome-wide (new Fig. 6e; APA score 10.16 vs 8.52 in siNC and siAS1 respectively, see Response Fig. 5). Additionally, CTCF looping strength decreased across all chromosomes individually (new Fig. 6f, see Response Fig. 5), consistent with ATAC-seq footprinting analysis in VCaP cells after knocking down IGF1R-AS1 (revised supplementary Fig. 6f, see Response Fig. 5). These data confirmed that IGF1R-AS1 positively affects CTCF looping genome-wide. Besides, in our analysis, we showed notably disrupted CTCF binding and reduced CTCF-mediated chromatin interactions around the E3 region in VCaP cells after the treatment with SWI/SNF degrader (Fig. 6b, see

Response Fig. 4). And we further showed that knocking down IGF1R-AS1 reduced the binding of SMARCA4, SMARCA1, CTCF, and the cohesin subunit SMC1A at the E3 enhancer (Fig. 6c, see **Response Fig. 5**). Moreover, we revealed that IGF1R-AS1 binds SMARCA4 through nucleotides 512-645 (new Fig. 4i, see **Response Fig. 3**), binds CTCF through nucleotides 177-407 (new Fig. 6h, see **Response Fig. 4**), and knocking down IGF1R-AS1 decreased the interaction between SMARCA4 and CTCF (new Fig. 6i, see **Response Fig. 4**). All together, these data indicate that IGF1R-AS1 acts as a scaffold to enhance interaction between SMARCA4 and CTCF to regulate chromatin looping and the epigenome.

We have added these results into **revised Fig. 6 and supplementary Fig. 6 (lines 441-446)** (see **Response Fig.5** for reference), and the proposed mechanism is summarized in a schematic model in **Fig. 6j** (see **Response Fig.5**).

6. The development of new targeting drugs based on this lncRNA, with thorough in vivo assessments of identity, PK, toxicity and efficacy, should improve the translational impact of this study.

Response: We sincerely appreciate the reviewer's insightful suggestion to develop novel IGF1R-AS1-targeting drugs to enhance the translational impact of our findings.

Our current study successfully establishes the vital role of IGF1R-AS1 in tumor progression and demonstrates its potential as a therapeutic target for PCa treatment. However, detailed drug development, including thorough PK, toxicity, and efficacy testing, is a complex process beyond the current scope of this foundational mechanistic manuscript. In our follow-up study, we will collaborate with RNA therapy experts to design and synthesize next-generation nucleic acid therapeutics (such as modified siRNAs or antisense oligonucleotides (ASOs)) targeting IGF1R-AS1. Furthermore, we will

generate a cell-based luciferase reporter high-throughput platform to screen small molecule inhibitors against IGF1R-AS1. Once we have the lead compounds, siRNAs, or ASOs, we will evaluate them in mouse models *in vivo* to assess their pharmacokinetics, safety profiles, and efficacy in inhibiting CRPC progression.

7. *Since IGF1R-AS1 was identified in metastatic castration-resistant prostate cancer (mCRPC) datasets, its potential role in resistance to AR-targeting therapies warrants investigation. In vivo assays addressing this question could increase the clinical importance of this study.*

Response: We thank the reviewer for this excellent advice.

Enzalutamide is a nonsteroidal antiandrogen for the treatment of mCRPC, both before and after chemotherapy. 22RV1 is a well-characterized enzalutamide-resistant prostate cancer cell line that expresses both AR and AR splice variants, including ARv7⁵. In 22RV1 cells, we depleted IGF1R-AS1 expression by two different siRNAs and/or shRNAs and performed *in vitro* and *in vivo* assays, demonstrating that knocking down IGF1R-AS1 inhibited 22RV1 cell growth, migration, invasion, and 22RV1 xenograft growth (**revised supplementary Fig. 3**, see **Response Fig. 2**). These results in 22RV1 cells demonstrated the potential to overcome enzalutamide resistance by targeting IGF1R-AS1 in prostate cancer treatment.

Reviewer #1 (Remarks on code availability):

N/A

Reviewer #2, expertise in lncRNAs, prostate cancer genomics and metastasis (Remarks to the Author):

The authors discover the antisense lncRNA IGF1R-AS1, transcribed from a high-density super-enhancer within the IGF1R locus, and delineate its oncogenic mechanism in metastatic castration-resistant prostate cancer (mCRPC). IGF1R-AS1 recruits SWI/SNF chromatin-remodelling and CTCF/cohesin architectural complexes to establish long-range loops that juxtapose distal MYC enhancers with the MYC promoter, thereby amplifying MYC expression and tumourigenicity. The work illustrates how a super-enhancer-driven lncRNA can remodel three-dimensional genome topology in trans to activate distal oncogenic circuitry in a cancer-specific context.

Response: We thank the reviewer for the excellent summary of the findings in our manuscript.

1. Threshold for lncRNA inclusion (Fig. 1a).

The manuscript states that lncRNAs retained for downstream analysis are those expressed in ≥ 5 of 101 mCRPC specimens. Please justify this 5-sample threshold. Does it represent a statistical cut-off (e.g. to control the false-discovery rate) or a biological rationale (e.g. prevalence $\geq 5\%$)?

Response: We thank the reviewer for this important question regarding our threshold for lncRNA retention. The threshold of ≥ 5 samples (representing $\sim 5\%$ prevalence) was chosen based on biological rationale rather than as a statistical false-discovery rate control.

Our reasoning was as follows:

- 1) Biological relevance: We aimed to focus on lncRNAs that are recurrently expressed across multiple patients, as these are more likely to represent biologically relevant features of mCRPC biology rather than patient-specific events or technical artifacts. lncRNAs expressed in $< 5\%$ of samples would be challenging to validate and are unlikely to represent generalizable disease mechanisms amenable to cohort-level analysis.
- 2) Balance between discovery and robustness: This threshold balances the goal of discovering potentially rare but recurrent lncRNAs while filtering sporadically detected transcripts that may reflect technical noise. Given the known low expression levels and high tissue-specificity of many lncRNAs, a stringent threshold could eliminate biologically relevant transcripts, while an overly permissive threshold would retain noise.
- 3) Practical considerations: Prevalence-based filtering is a common approach in large-scale cancer transcriptome analyses to focus on consistently detectable features.

To address the reviewer's concern, we performed sample threshold analysis for lncRNA discovery in these 101 mCRPC specimens, showing that when setting different thresholds, the number of known prostate cancer relevant lncRNAs (including PCA3, SCHLAP1, MALAT1, HOTAIR, PCAT1, PVT1, UCA1, SNHG20, HOTTIP, PCGEM1, CTBP1-AS, ARLNC1, MIR503HG, DANCR, NEAT1, TUG1, LINC01128, PCAT1, PCAT5, PCAT7, PCAT14, PCAT18, PCAT19, SNHG3, SNHG12, SOCS2-AS1, TRPM2-AS, MEG3) retained changes accordingly. This analysis demonstrates that at a threshold of ≥ 5 samples, $\sim 90\%$ of known lncRNAs are retained, supporting our choice as appropriately balanced between sensitivity and specificity (**Response Fig. 6**). We have revised the Methods section entitled 'Novel lncRNA identification'

to explicitly justify the prevalence threshold, clarifying that transcripts detected in fewer than five patient samples were excluded to enrich for recurrently expressed lncRNAs and minimize sporadic, noise-driven detections (line 540-542).

2. Please describe explicitly how many of the 101 tumors express IGF1R-AS1 above background and provide the expression distribution (e.g. violin plot or histogram).

Response: We thank the reviewer for this important suggestion. We reanalyzed IGF1R-AS1 expression across primary PCa (n=500) and mCRPC (n=101) samples based on the IGF1R-AS1 feature junction (chr15: 98839896-98843675). We found that 33 samples (32.7%) in 101 mCRPC, and 89 samples (17.8%) in 500 primary PCa express IGF1R-AS1 with its feature junction.

We have now added a supplementary Fig. 3a (lines 239-240) (see **Response Fig. 7**) showing the expression distribution among PCa samples.

3. Most functional assays are performed in VCaP cells. To strengthen the claim that IGF1R-AS1 is broadly oncogenic in mCRPC, please test at least one additional metastatic line with high endogenous expression (e.g., MDA-PCa-2b or 22Rv1) using knockdown. Conversely, introduce IGF1R-AS1 into a line (e.g., LNCaP) to determine whether overexpression is sufficient to confer the oncogenic phenotypes.

Response: We thank the reviewer for providing these critical suggestions.

We performed functional characterization of IGF1R-AS1 in a second PCa cell line, 22RV1 which is a CRPC cell line expressing AR wild-type and AR-v7 variant (see **Response Fig. 8**). We performed IGF1R-AS1 knockdown using two independent siRNAs or shRNAs in 22RV1 cells, and then evaluated cell growth, migration, and invasion. Consistent with the findings in VCaP cells, knocking down IGF1R-AS1 in 22RV1 cells suppressed cell growth, migration, and invasion (**new Supplementary Fig. 3b-d, lines 242-246**). Furthermore, we performed *in vivo* xenograft assays using 22RV1 cells with IGF1R-AS1 stable knockdown by two different shRNAs (shAS1-1# and shAS1-2#), which also showed inhibited tumor progression *in vivo* (**new Supplementary Fig. 3k-n, lines 252-253**). In addition, we stably overexpressed IGF1R-AS1 in LNCaP cells using lentivirus, resulting in modest increases in cell growth, migration, and invasion (**new Supplementary Fig. 3h-j, lines 248-249**), further demonstrating the tumor progression role of IGF1R-AS1 in prostate cancer.

4. Selection of TCGA specimens (Fig. 2b). Only three “positive” TCGA prostate samples are shown. Clarify the selection criteria (“positive” is ambiguous). Were these the top three expressers of IGF1R-AS1, or the only cases with matched normal tissue?

Response: We thank the reviewer for this valuable comment. In Fig. 3b, the three “positive” TCGA prostate cancer samples correspond to tumors exhibiting the highest IGF1R-AS1 expression among cases with available matched normal tissue. These samples were chosen to illustrate representative expression patterns in tumors with elevated IGF1R-AS1 levels. We have revised the description in the manuscript as follows: “We examined **the top three IGF1R-AS1-high** TCGA prostate specimens for which tumor and matched adjacent normal tissue RNA-seq data were available” (**line 234**).

5. The author described that both SMARCA1 and SMARCA4 interact with the MYC promoter and enhancer regions and bind IGF1R-AS1. However, the subsequent analyses rely exclusively on SMARCA4 ChIP-seq data. Are the authors focusing solely on SMARCA4 because SMARCA1 and SMARCA4 operate within the same SWI/SNF chromatin-remodeling complex? Please clarify this point in the manuscript.

Response: We appreciate the reviewer's careful reading and agree that our focus on SMARCA4 in Figure 5 requires clarification.

First, we wish to clarify that SMARCA1 and SMARCA4 operate in distinct chromatin remodeling complexes with different functions. SMARCA4 is the catalytic ATPase subunit of the SWI/SNF complex, while SMARCA1 is the catalytic ATPase of the ISWI complex. These complexes have complementary but distinct roles in chromatin regulation: SWI/SNF complexes typically promote chromatin accessibility and gene activation, whereas ISWI complexes primarily regulate nucleosome spacing and positioning, though their effects on gene expression can be context-dependent⁶.

We focused our chromatin accessibility and network analyses on SMARCA4 and the SWI/SNF complex for several reasons:

- 1) **Biological relevance:** SWI/SNF plays a critical role in prostate cancer biology and has emerged as a therapeutic target in this disease⁷. Many drugs have been developed to pharmacologically suppress SWI/SNF function in cancer models, including testing in prostate cancer models, such as ADAADi, PB16, AU15330, I-78, 2-77, and some of these drugs are now being evaluated in clinical trials for cancer treatment⁸.
- 2) **Functional alignment:** SWI/SNF's role in promoting chromatin accessibility aligns with our observations of IGF1R-AS1-dependent accessibility changes.
- 3) **Data availability:** High-quality SMARCA4 ChIP-seq data in VCaP cells were publicly available, enabling genome-wide identification of direct SWI/SNF binding sites. This was essential for distinguishing IGF1R-AS1-regulated regions that are directly bound by chromatin remodelers from those regulated through secondary mechanisms.

Additionally, we performed complementary network analysis using ATAC-seq data from SMARCA1 knockdown cells (**new Supplementary Figure 5g**, see **Response Fig. 9**). While MYC was not the top dysregulated gene in SMARCA1 network as in SMARCA4 (**Supplementary Figure 5f**, see **Response Fig. 9**), it was identified among the top 40 hub genes in the SMARCA1 network (**new Supplementary Figure 5g**, see **Response Fig. 9**), consistent with our findings that both remodelers regulate MYC (**Fig. 4n-q**, **Supplementary Figure 4 i-l**, see **Response Fig. 9**, **lines 350-352**) and interact with IGF1R-AS1.

6. Minor comments

Typographical error (Fig. 2h). “siNON” should read “siCon” (scramble control).

Labelling ambiguity (Fig. 3h). The axis label “siAS-1 #1/#2 meta” is unclear. If both siRNAs were co-transfected, please indicate “mix” or “pool”.

Response: We thank the reviewer for your valuable comment. We apologize for the error and ambiguity. We have corrected “siNON” to “siCon” in Fig. 2h. In Fig. 3j, the label “siAS-1 #1/#2 meta” refers to a gene set enrichment analysis (GSEA) performed on the results of a meta-analysis of differentially expressed genes (DEGs) from independent siAS-1 #1 and siAS-1 #2 knockdown experiments. The two siRNAs were not co-transfected; rather, DEG analyses were conducted separately and then combined via meta-analysis prior to GSEA. We have revised the figure legend accordingly to clarify this point.

Reviewer #3, ECR (Remarks to the Author):

Response: We sincerely appreciate the reviewer's evaluation of our manuscript.

Reviewer #4, expertise in non-coding RNA and transcriptional regulation, super-enhancers, epigenetics and cancer (Remarks to the Author):

In this manuscript, Yang et al. identify IGF1R-AS1, a previously uncharacterized tumor-specific lncRNA, as a key regulator in metastatic castration-resistant prostate cancer (mCRPC). Through integrative multi-omics analyses, the authors demonstrate that IGF1R-AS1 interacts with the chromatin remodelers SMARCA1 and SMARCA4 to orchestrate long-range chromatin looping between the distal MYC enhancer E3 and its promoter, thereby amplifying MYC expression and driving tumorigenesis. This work not only unveils a novel trans-acting mechanism by which lncRNAs modulate oncogenic signaling but also highlights the broader potential of tumor-specific lncRNAs as therapeutic targets in cancer. This is a well-executed study with significant findings. However, some mechanistic details require further refinement before publication. Below is a detailed review with suggestions for further improvement:

Response: We thank the reviewer for the supportive comments.

1. The manuscript suggests IGF1R-AS1 recruits SWI/SNF to MYC enhancers, but the exact binding sites or motifs mediating this interaction are unclear. Additional experiments (e.g., truncation mutants of IGF1R-AS1 to identify functional domains) could strengthen this claim. Moreover, it is interesting to see whether IGF1R-AS1 directly binds SMARCA1 and SMARCA4. Can these two proteins directly bind to IGF1R-AS1 in vitro? If so, which functional domains are responsible for the direct interaction?

Response: We thank the reviewer for the excellent suggestions.

First, to investigate whether IGF1R-AS1 directly interacts with SMARCA1 and SMARCA4, we performed *in vitro* RNA-protein binding assay using *in vitro*-transcribed biotin-labeled IGF1R-AS1 RNA and recombinant His-tagged SMARCA4 protein, or Flag-tagged SMARCA1 protein, which demonstrated that IGF1R-AS1 binds directly to SMARCA1 protein and SMARCA4 protein (new Fig. 4f-g).

To map the IGF1R-AS1 functional motifs corresponding to SMARCA1 and SMARCA4 binding, we analyzed the secondary structure of IGF1R-AS1 (new Fig. 4h), and conducted *in vitro* RNA pull-down assays using a series of

truncated biotin-labeled IGF1R-AS1 fragments (**new Fig. 4i**). This analysis revealed that nucleotides 512-645 of IGF1R-AS1 are critical for the interaction with SMARCA4 protein, while nucleotides 177-407 and 408-511 of IGF1R-AS1 are essential for the interaction with SMARCA1 protein.

Both SMARCA1 and SMARCA4 contain an ATPase domain in the middle of the protein sequence (**new Fig. 4j-k**). We cloned the full-length, N-terminal domain, ATPase domain, and C-terminal domain of both SMARCA1 and SMARCA4, each with a C-terminal Flag tag. We expressed them in VCaP cells and then performed RIP assay using anti-Flag antibody (**new supplementary Fig. 4g-h**), followed by qPCR to analyze the enrichment of IGF1R-AS1 by different protein domains (**new Fig. 4j-k**). The results showed that IGF1R-AS1 mainly binds to the ATPase domain of SMARCA4 and the C-terminal domain of SMARCA1 (**new Fig. 4j-k**).

These detailed mapping studies significantly strengthen our findings by identifying the specific RNA motifs and protein domains responsible for the IGF1R-AS1-SMARCA4 and IGF1R-AS1-SMARCA1 interaction. These new data have been incorporated into **revised Fig. 4 (lines 292-302)** (and shown here in **Response Fig. 3** for reference).

2. As *CCAT1* is localized in a super-enhancer region, it is extensively transcribed in many types of cancers and has been demonstrated to have a 5' additional extension in HeLa cells (see Cai et al., Nature, 2020), I am wondering whether IGF1R-AS1 RNA directly interact with the enhancer RNAs transcribed in E3 region? could they form RNA base pairings? This can explain why IGF1R-AS1 specifically targets the chromatin remodeling complex to this enhancer and potentially to thousands of other sites.

Response: We thank the reviewer for this insightful suggestion.

First, we analyzed RNA-seq data from prostate cancer cell lines, including VCaP and 22Rv1, both of which express IGF1R-AS1. We found that neither the canonical *CCAT1* nor its long isoforms are expressed in these prostate cancer cells (**revised Supplementary Fig. 6a**; see **Response Fig. 10**). In contrast, analysis of RNA-seq data from HCT116, a colon cancer cell line known to express high levels of the long *CCAT1* isoform⁹, confirmed robust *CCAT1* expression and the presence of a super-enhancer marked by strong H3K27ac ChIP-seq signals in this region (**revised Supplementary Fig. 6a**; see **Response Fig. 10**). We further examined RNA-seq data from tumor tissue samples in the TCGA cohort, which also showed low *CCAT1* expression in prostate cancer (**revised Supplementary Fig. 6b**; see **Response Fig. 10**). Collectively,

these results indicate that CCAT1 or its long isoforms, is unlikely to participate in IGF1R-AS1-mediated regulation in the context of our study.

In our study, we found that IGF1R-AS1 interacts with SMARCA4 and CTCF through different nucleotide regions (**new Fig. 4h-i**, see **Response Fig. 3** and **new Fig. 6h**, see below **Response Fig. 4**), and knocking down IGF1R-AS1 reduced the interaction between SMARCA4 and CTCF (**new Fig. 6i**, see below **Response Fig. 4**), which led to disrupted CTCF binding and reduced CTCF-mediated chromatin interactions around the E3 region in VCaP cells, as well as reduction in CTCF looping strength genome-wide (**new Fig. 6c-f**, see below **Response Fig. 5**). These data showed that IGF1R-AS1 acts as a scaffold to promote the interaction between SMARCA4 and CTCF at this enhancer and potentially thousands of other sites to regulate gene expression.

3. While the study shows *IGF1R-AS1* does not regulate *IGF1R*, it would be helpful to explicitly test whether *IGF1R-AS1* knockdown affects other nearby genes (e.g., by RNA-seq or qPCR).

Response: We thank the reviewer for this great advice.

We performed qPCR to assess the expression of two IGF1R-neighboring genes, SYNM and TTC23, which are expressed in VCaP cells. We found that their transcription was not significantly affected by knocking down IGF1R-AS1 in VCaP, 22RV1, and H727 cells (revised Fig. 3I and Supplementary Fig. 3p-q, line 266, see below **Response Fig. 11**). These results further support that IGF1R-AS1 functions *in trans*, rather than *in cis*, in regulating gene expression.

4. For the IGF1R-AS1 knockdown studies, it is better to reintroduce IGF1R-AS1 in knockdown cells (e.g., via overexpression) to confirm that the observed phenotypes are due to IGF1R-AS1 loss and not off-target effects.

Response: We thank the reviewer for this important suggestion. We reintroduced IGF1R-AS1 by transfecting an IGF1R-AS1 overexpression construct into VCaP IGF1R-AS1 knockdown cells and observed rescued cell growth, migration, and invasion (new Supplementary Fig. 3e-g, lines 247-248), and a rebound in Myc expression (new Supplementary Fig. 3t-u, lines 267-268), confirming that the observed phenotypes are due to IGF1R-AS1 loss but not off-target effects. See below Response Fig. 12.

5. The Tempus cohort analysis suggests IGF1R-AS1 activity predicts poor survival. Expanding this to other cohorts or validating in independent datasets would strengthen the biomarker claim.

Response: We thank the reviewer for this excellent suggestion.

We have now expanded our survival analysis to include two independent patient cohorts (new supplementary Fig. 5l-m, see Response Fig. 13), thereby substantially strengthening the clinical relevance of IGF1R-AS1 as a prognostic biomarker.

We first analyzed survival data from a microarray based study in primary prostate cancer (Taylor et al., 2010¹⁰; n=140) and found that high IGF1R-AS1 ATAC-seq based network activity score predicts significantly worse disease-free survival (CoxPH p = 0.0135; new Supplementary Figure 5l). We further validated these findings in an independent mCRPC cohort from the SU2C study (Abida et al., 2019¹¹; n=71), where IGF1R-AS1-high patients showed significantly worse overall survival compared to IGF1R-AS1-low patients (CoxPH p = 0.0146; new Supplementary Figure 5m). This independently confirms our findings from the Tempus mCRPC cohort in a separate patient population. Importantly, we found that IGF1R-AS1 stratifies patient outcomes even among MYC-high patients. In the SU2C cohort, patients with both high IGF1R-AS1 and high MYC expression had significantly worse survival than MYC-high patients with low IGF1R-AS1 (p = 0.0015; new Supplementary Figure 5m). This demonstrates that IGF1R-AS1 provides independent prognostic information beyond MYC expression alone, consistent with our mechanistic model in which IGF1R-AS1 amplifies MYC-driven oncogenesis through regulation of chromatin architecture. To further assess IGF1R-AS1's clinical utility, we compared its prognostic performance to the expression of Prolaris cell cycle progression markers, which are utilized in an FDA-cleared test for prostate cancer risk stratification. IGF1R-AS1 network score demonstrated comparable or superior prognostic performance to Prolaris markers z-score as measured by concordance index (new Supplementary Figures 5m).

We have revised the Results section (lines 375-379) to describe the multi-cohort validation and added new Supplementary Figures 5l-m showing the survival analyses in the Taylor and SU2C cohorts.

6. Several typos need to be fixed: Page 3: "evolutionally" → "evolutionarily"; Page 9: "fueling cancer progression" → "driving cancer progression"

Response: We apologize for these errors (lines 130 and 516). We have corrected them and carefully checked the typos throughout the manuscript.

Reference

- 1 Li, Y. *et al.* A Genomic Language Model for Chimera Artifact Detection in Nanopore Direct RNA Sequencing. *bioRxiv* (2024). <https://doi.org/10.1101/2024.10.23.619929>
- 2 Valletta, M. *et al.* Exploring the Interaction between the SWI/SNF Chromatin Remodeling Complex and the Zinc Finger Factor CTCF. *Int J Mol Sci* **21** (2020). <https://doi.org/10.3390/ijms21238950>
- 3 Marino, M. M. *et al.* Interactome mapping defines BRG1, a component of the SWI/SNF chromatin remodeling complex, as a new partner of the transcriptional regulator CTCF. *J Biol Chem* **294**, 861-873 (2019). <https://doi.org/10.1074/jbc.RA118.004882>
- 4 Michel, B. C. *et al.* A non-canonical SWI/SNF complex is a synthetic lethal target in cancers driven by BAF complex perturbation. *Nat Cell Biol* **20**, 1410-1420 (2018). <https://doi.org/10.1038/s41556-018-0221-1>
- 5 Dehm, S. M., Schmidt, L. J., Heemers, H. V., Vessella, R. L. & Tindall, D. J. Splicing of a novel androgen receptor exon generates a constitutively active androgen receptor that mediates prostate cancer therapy resistance. *Cancer Res* **68**, 5469-5477 (2008). <https://doi.org/10.1158/0008-5472.CAN-08-0594>
- 6 Barisic, D., Stadler, M. B., Iurlaro, M. & Schubeler, D. Mammalian ISWI and SWI/SNF selectively mediate binding of distinct transcription factors. *Nature* **569**, 136-140 (2019). <https://doi.org/10.1038/s41586-019-1115-5>
- 7 Xiao, L. *et al.* Targeting SWI/SNF ATPases in enhancer-addicted prostate cancer. *Nature* **601**, 434-439 (2022). <https://doi.org/10.1038/s41586-021-04246-z>
- 8 Dreier, M. R., Walia, J. & de la Serna, I. L. Targeting SWI/SNF Complexes in Cancer: Pharmacological Approaches and Implications. *Epigenomes* **8** (2024). <https://doi.org/10.3390/epigenomes8010007>

- 9 Xiang, J. F. *et al.* Human colorectal cancer-specific CCAT1-L lncRNA regulates long-range chromatin interactions at the MYC locus. *Cell Res* **24**, 513-531 (2014). <https://doi.org/10.1038/cr.2014.35>
- 10 Taylor, B. S. *et al.* Integrative genomic profiling of human prostate cancer. *Cancer Cell* **18**, 11-22 (2010). <https://doi.org/10.1016/j.ccr.2010.05.026>
- 11 Abida, W. *et al.* Genomic correlates of clinical outcome in advanced prostate cancer. *Proc Natl Acad Sci U S A* **116**, 11428-11436 (2019). <https://doi.org/10.1073/pnas.1902651116>

Point-by-Point response

RE: **NCOMMS-25-35735A**- Tumor-specific lncRNA *IGF1R-ASI* trans-regulates chromatin interactions associated with oncogenic MYC signaling

We would like to thank the editor for the opportunity to allow us to further revise our manuscript and thank all reviewers for your insightful and supportive comments, which have significantly improved the quality of our work. As requested, we have rewritten a few parts of the manuscript. We believe that these changes have resulted in a more solid and cohesive manuscript. We will now provide a point-by-point response to the reviewers' comments. We hope the Reviewers and the editor will find this manuscript to be improved and suitable for publication.

REVIEWER COMMENTS

Reviewer #1 (Remarks to the Author):

The authors have comprehensively addressed my concerns. The incorporation of Nanopore direct RNA sequencing, the validation in a second CRPC cell line (22RV1), the detailed domain mapping of the lncRNA-protein interactions, and the CTCF HiChIP analysis strengthened the novelty, mechanistic depth, and robustness of the study. I have no further comments.

Response: We are sincerely grateful for the reviewer's positive feedback.

Reviewer #2 (Remarks to the Author):

Overall, the authors' responses are clear, well-reasoned, and comprehensive, and the issues raised have been well addressed. I have minor comments for the paper.

Response: We greatly appreciate the reviewer's supportive comment.

1. Introduction: 'Opening new avenues for therapeutic intervention' feels too ambitious given the current data. The study compellingly demonstrates tumor-specific lncRNA-mediated distal regulation and 3D chromatin architecture, but the evidence does not yet support clinical or therapeutic claims.

Response: We thank the reviewer for this comment. We removed the phrase "opening new avenues for therapeutic intervention" from the introduction.

2. Response 7. The current distribution plot is not straightforward for evaluating prevalence. The authors' point is that the IGF1R-ASI junction is more prevalent in metastatic vs localized tumors, showing raw sample counts obscure the comparison because the metastatic (Met) and localized (PCa) cohorts differ in size. A pie chart summarizing the proportions should be sufficient.

Response: We thank the reviewer for this suggestion. We retained only the pie chart and removed the bar plot from the figure, making it straightforward to evaluate the distribution of the *IGF1R-AS1* junction in PCa patient samples. See revised Supplementary Figure 3a below.

3. Following the authors' clarification of the rationale for studying *SMARCA4*, I recommend inserting concise biological relevance and functional alignment justifications to strengthen clarity and coherence.

Response: We thank the reviewer for this great suggestion. We added the justification in the discussion: “Due to the SWI/SNF complex’s critical role in PCa biology and *SMARCA4*’s alignment with *IGF1R-AS1* in the regulation of chromatin accessibility, we focused on investigating *SMARCA4*’s participation in *IGF1R-AS1*-regulated *MYC* promoter-enhancer interaction.” (line 489-492)

Reviewer #3 (Remarks to the Author):

Response: We greatly appreciate the reviewer's effort in reviewing our manuscript.

Reviewer #4 (Remarks to the Author):

The revised manuscript has fully addressed all of my concerns, and I am pleased to support its publication.

Response: We are grateful that the reviewer supports the publication of our manuscript.